



# Three-Dimensional Reconstruction of Ocean Circulation from Coastal Marine Observations: Challenges and Methods

Ivan Manso-Narvarte[1], Erick Fredj[2], Gabriel Jordà[3], Maristella Berta[4], Annalisa Griffa[4], Ainhoa
Caballero[1] and Anna Rubio[1]

[1] AZTI-Marine Research, Pasaia, Spain
[2] Department of Computer Sciences, Jerusalem College of technology, Jerusalem, Israel
[3] Instituto Español de Oceanografía, Centre Oceanogràfic de Balears, Palma de Mallorca, Spain
[4] ISMAR, CNR, La Spezia, Italy

*Correspondence to*: Ivan Manso-Narvarte (imanso@azti.es)

**Abstract.** Monitoring and investigating the dynamics of coastal currents is crucial for the development of environmentally
sustainable coastal activities, in order to preserve marine ecosystems as well as to support marine and navigation safety. This
need is driving the set-up of a growing number of multiplatform operational observing systems, aiming to the continuous
monitoring of the coastal ocean. A significant percent of the existing observatories is today equipped with land-based High
Frequency Radars (HFR), which provide real-time currents with unprecedent coverage and resolution, limited however, to the
surface layer. The combination of data from HFR with complementary data from in-situ platforms providing information of
the currents at subsurface layers (ADCP moorings) is investigated here to reconstruct the 3D current velocity field from in-
situ observations. For this purpose, two methods based on different approaches are used. On the one hand, the Reduced Order
Optimal Interpolation which is fed, in this case, with a spatial covariance matrix extracted from a realistic numerical oceanic
simulation; and on the other hand, the Discrete Cosine Transform Penalized Least Square, which is a data gap-filling method
based on penalized least squares regression that balances fidelity to the data and smoothness of the solution.

As a proof of concept, we test the methods' skills by using emulated observations of currents, extracted from a numerical
simulation (3D reference field). The test set-up emulates the real observatory scenario in the study area (south-eastern Bay of
Biscay), which includes a long-range HFR and two ADCP moorings inside the HFR footprint area. Then, the reconstructed
fields (outputs of the methods) are compared with the 3D reference fields. In general, the results show satisfactory 3D
reconstructions with mean spatial (for each depth level) errors between 0.55–10.94 cm s$^{-1}$ for the first 150 m depth. The
methods perform better in well sampled areas, and although different performances between the methods are observed, both
show promising skills for the computation of new operational products integrating complementary observations, broadening
the applications of in-situ observational data.



## 1 Introduction

Multiplatform observing systems are arising in several areas of the coast for providing data at different spatio-temporal scales. The combination of such data is a powerful approach for a better monitorization and understanding of the 3D coastal circulation, which is a key aspect to support marine and navigation safety, as well as to preserve marine ecosystems.

Among the different observing systems, High Frequency Radar (HFR) technology offers a unique insight into coastal ocean variability, by providing information of the ocean and atmosphere interface for a better understanding of the coupled ocean-atmosphere system and the surface ocean coastal dynamics. In addition, since HFR data provide measurements of currents with a relatively wide spatial coverage (up to 200 km from the coast) and high spatial and temporal resolution (typically a few km and one hour) in near real time, they have become invaluable tools in the field of operational oceanography. Recent reviews

on this technology and its applications worldwide have been provided by several authors (Fuji et al., 2013; Paduan and Washburn, 2013; Wyatt, 2014, Rubio et al., 2017; Roarty et al., 2019). However, HFRs provide current data only relative to the surface within an integration depth ranging from tens of cm to 1–2 m, depending on the operating frequency (see Rubio et al., 2017). Moreover, data coverage is not always regular and contain spatial and temporal data gaps due to several environmental, electromagnetic and geometric causes (Chapman et al., 1997).

The combination of HFR data with complementary data of coastal currents in the water column is especially useful since it enables to increase the temporal and spatial coverage and expand the information towards subsurface layers, broadening their application to biological, geochemical and environmental issues, since plankton or pollutants can be located deeper in the water column and not only follow surface dynamics. Nevertheless, the combination of independent measurements of the ocean surface currents with those along the water column is challenging since the surface and the water column dynamics may

respond to different forcing and can be characterized by different space and time scales. Moreover, the measurements at the surface and in the water column are done under different observation principles and may have different space and time coverage and resolution.

Several data gap-filling/reconstruction methods such as the ones in Table 1 have been widely used in different studies applied to oceanographic data sets (e.g. Yaremchuk and Sentchev, 2009; Hernández-Carrasco et al., 2018; Taillandier et al., 2006;

Jordà et al., 2016; Fredj et al., 2016; Esnaola et al., 2013; Alvera-Azcárate et al., 2007; Barth et al., 2014). In this work we aim to explore the use of two of them for the 3D reconstruction of high-resolution coastal current fields by combining information from one long-range HFR and two moorings equipped with ADCPs providing data every 8 meters from the surface to ~150 m depth inside the HFR footprint area. Hence, the skills of two data-reconstruction methods are assessed and compared, aiming to give a first step towards their applicability for this specific case. These two methods were chosen because of their good

performances in previous attempts for the reconstruction of HFR current data and because they rely on different basic principles: the Discrete Cosine Transform Penalized Least Square (DCT-PLS), implemented by Fredj et al. (2016), is based on the fitting of a function, and the Reduced Order Optimal Interpolation (ROOI), implemented by Jordà et al. (2016), uses an approximation to the velocity covariances. The assessment of the performances of both methods is carried out in terms of





current velocities, using a scenario based on a real observatory located at the south eastern Bay of Biscay (SE-BoB) (Fig. 1a). To that aim, a synthetic reality experiment is performed. In particular, the outputs of a realistic numerical simulation in the study area are used as synthetic reality from which observations are extracted (emulating data from one HFR and two ADCPs moored inside the HFR footprint area). The results are then compared to the original numerical simulation outputs (reference

field) to evaluate the quality of the reconstructed fields and quantify the methods' skills.

## 2 Methods and data

### 2.1 Study area and main approach

The study area is located in the SE-BoB, which is characterized by the presence of canyons (e.g. Capbreton canyon), by an abrupt change in the orientation of the coast and by a narrow shelf and slope (see Fig. 1). The winter surface circulation in the

SE-BoB is mainly related to a slope current flowing, in the upper 300 m of the water column, eastwards along the Spanish coast and northwards along the French coast (the so-called Iberian Poleward Current, IPC) (Frouin et al., 1990; Haynes and Barton, 1990; Pingree and Le Cann, 1990, 1992a, 1992b; Peliz et al., 2003; Le Cann and Serprette, 2009) with maximum currents of 70 cm s$^{-1}$ (Solabarrieta et al., 2014). In summer, the flow is reversed being three times weaker than in winter (Solabarrieta et al., 2014). In the water column, the sub-surface properties measured by two slope buoys show a seasonal

variability (Rubio et al., 2013). Whilst in winter, the water column is well-mixed and shows stronger currents (strongest currents ranging from 20 cm s$^{-1}$ to 50 cm s$^{-1}$), in summer, it is stratified with mean thermocline depths ranging from -30 to -50 m, with temperatures over 20 °C and with weaker currents (strongest currents ranging from 10 cm s$^{-1}$ to 20 cm s$^{-1}$).

The multiplatform observing system available in this area belongs to the Basque Operational Observing System (EuskOOS, www.euskoos.eus). For this study, two current vertical profiles emulating the data obtained by ADCPs located in two slope

buoys (Matxitxako and Donostia buoys) along the Spanish coast, and surface currents fields emulating the fields obtained by a HFR were extracted from a realistic numerical simulation. The emulated HFR coverage area and the locations of the emulated slope buoys are shown in Fig. 1b.

Since the current regime is seasonally modulated, the performances of the two data-reconstruction methods were tested for winter and summer periods: Nov-Dec-Jan-Feb (2010-2011) and Jun-Jul-Aug-Sep (2011), respectively.

The data-reconstruction methods were also analysed in a reduced grid case to evaluate the performance of the reconstructions in areas of highly correlated currents to the observations. Note that in this study these areas generally correspond to the closest points to the observations, therefore, hereinafter these areas will be called well sampled areas. This reduced grid was determined by the area where the reference fields' surface zonal current velocity component (U) temporal cross-correlation values, between the ADCP locations and the rest of the grid, are higher or equal to 0.8. Note that this grid is slightly different

for each season, however, it mainly covers the Spanish slope area (grids delimited by black and orange lines in Fig. 1b). The meridional current velocity component (V) was not considered for determining this grid since U is the velocity component that dominates the current regime at the nearby areas of the ADCP locations.

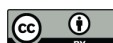



Moreover, a second scenario with two additional current vertical profiles (4-buoy scenario) along the French slope (see Fig. 1b) was also considered in order to assess the sensitivity of the methods to different observational configurations.

The approach used for the analysis of the reconstructions is as follows (see Fig. 2): first, the outputs of a numerical simulation were used to extract the observations that emulate the data obtained from a real coastal observatory scenario. These
observations were then used as inputs for the data-reconstruction methods. Note that, in addition, the ROOI uses historical data as input to define spatial covariances. Finally, the 3D reconstructed fields were compared to the original outputs of the numerical simulation (reference field) to assess the performances of the data-reconstruction methods.

## 2.2 Data-reconstruction methods

### 2.2.1 ROOI method

The ROOI was first proposed by Kaplan et al. (1997) to reconstruct sea surface temperatures (SST) from sparse data and has been applied since then for different variables such as sea level pressure (Kaplan et al., 2000), sea level anomalies (Church and White, 2006), or 3D velocity fields (Jordà et al., 2016). It is based on Empirical Orthogonal Function (EOF) decomposition and the details can be found in Kaplan et al. (1997, 2000) or Jordà et al. (2016), so here only the basic elements are presented. Expressing the 3D velocity field as a matrix $Z(r,t)$, where $r$ is the $m$-vector of spatial locations and $t$ the $n$-vector of times, a
spatial covariance matrix is first computed as $C = n^{-1}ZZ^T$. Then, an EOF decomposition can be applied:

$$C = U\Lambda U^T \tag{1}$$

where $U$ is an $m \times m$ matrix whose columns are the spatial modes (EOFs) and $\Lambda$ is the $m \times m$ diagonal matrix of eigenvalues. The velocity field can then be exactly reproduced as:

$$Z(r,t) = U(r) \cdot \alpha(t) \tag{2}$$

in which $\alpha$ can be computed as $\alpha = U^T Z$.

In practice, the velocities at every grid point of the 3D analysis grid are not known, but only at a limited set of $N$ locations, being usually $N << m$. The problem we intend to solve is precisely that of retrieving the whole matrix $Z$ from the available observations (e.g. surface velocities from HFR and velocity profiles at the ADCP locations). The first problem is that the eigenvector $U$ and eigenvalue $\Lambda$ matrices cannot be computed from actual observations (i.e. there are not enough samples), so
a common choice is to use historical data from a realistic numerical simulation to represent the actual velocity statistics. A second aspect to be considered is that fitting high order modes may introduce unwanted noise in the reconstruction. Thus, the Eq. (2) is truncated to include only the $M$ leading EOFs, so that the contribution of the higher-order modes (accounting for local, small-scale features) is neglected:

$$Z_M(r,t) = U_M(r) \cdot \alpha_M(t) \tag{3}$$

The next problem is that obviously the amplitudes cannot be obtained as in Eq. (2), since now we do not know $Z$. Instead, the $M$ amplitudes can be determined under the constraint that the reconstructed $Z_M$ fits the observations available at each time



step. More generally, the amplitudes are obtained minimizing a cost function that takes into account the observational noise and the role of neglected modes (see Kaplan et al., 1997, 2000, for the complete derivation).

Summarizing, using the ROOI, the values of the velocity at every grid point of a predefined 3D grid can be obtained merging the spatial modes of variability computed from a realistic numerical simulation (used as historical data) and the temporal

amplitudes obtained using the available observations. Several sensitivity tests have been performed to tune the method and finally 20 modes have been considered ($M$=20). Regarding the spatial modes of variability, they have been obtained from different numerical simulations (see Sect. 2.3) to test the sensitivity of the results to the accuracy in the definition of the spatial covariances.

### 2.2.2 DCT-PLS method

The DCT-PLS is a straightforward data gap-filling method proposed by García (2010), based on a penalized least square regression. Fredj et. al. (2016) has shown that the method is capable of filling data gaps in the HFR surface current networks along the mid-Atlantic coast of the United States with high accuracy. Here it is used for filling current data from emulated HFR and ADCP fields in a 3D grid in the SE-BoB. In this section the basic principle of the method is explained, however, for more details the reader is referred to García (2010) or Fredj et al. (2016).

The main aim of the method is to find the best fitting model, which is based on Discrete Cosine Transforms (DCTs) and one smoothing (fitting) parameter $s$. Thus, the fitting model that correspond to each $s$ is tested by cross-validation in order to obtain the best one. The general approach of the method is as follows: for each $s$ (i.e. for each fitting model) the observations are split into two subsets, the training set, which is used to fit the model, and the test set, which is used to test it. This test is carried out by the trade-off ($F$) between the bias of the fitting (residual sum of squares $RSS$) and the variance of the results of the created

model (penalty term $P$):

$$F = RSS + P = \|y - \hat{y}\|^2 + s\|D\hat{y}\|^2 \tag{4}$$

where $y$ is the data of the test set, $\hat{y}$ is the data of the created model and $D$ is a second order difference derivative. Then, for the same $s$, this procedure is repeated for different training and test sets obtaining different $F$ values at each time. The mean value of $F$ (that is, $E[F]$) will provide a General Cross Validation ($GCV$) score that correspond to each fitting model (i.e. to

each $s$):

$$E[F] \rightarrow GCV \tag{5}$$

, and the best fitting model will be the one that minimizes the $GCV$ score:

$$\min(GCV) \rightarrow s. \tag{6}$$

In conclusion, here we introduce a penalized least square method, based on 4D discrete cosine transforms, with one smoothing

parameter approach consisting of minimizing a criterion that balances the fidelity with the current data, measured by the $RSS$ and a $P$ that reflects the noisiness of the smooth current data.





### 2.3 Numerical simulations

The Atlantic-Iberian Biscay Irish simulation, and particularly the IBI_REANALYSIS_PHYS_005_002 product (hereinafter IBI), provided by the Copernicus Marine Environment Monitoring Service (CMEMS), was used to obtain the observations and the reference fields. The IBI reanalysis is based on a realistic configuration of the NEMO model for the Iberian Biscay

Irish region (Fig. 1a) that assimilates in situ and satellite data. For more details see Table 2 and a complete description about the product and its validation can be found in Sotillo et al. (2015) and the links shown in Table 2.

For the ROOI the definition of the spatial covariances is required and, in our case, it has been obtained from additional outputs from numerical simulations (see Fig. 2) with daily data from 1993 to 2009. In particular, in addition to IBI, two different numerical simulations were used for this purpose: the GLORYS high resolution (HR) product

(GLOBAL_REANALYSIS_PHY_001_030) (hereinafter GLORYS-HR), and a GLORYS low resolution (LR) product (GLOBAL_REANALYSIS_PHY_001_025) (hereinafter GLORYS-LR). The general details of these products are listed in Table 2 along with links to information about the products and their validation. The goal of using different simulations is to explore the impact of having an imperfect definition of the covariances. Thus, the ROOI method was tested both in an optimal configuration, where the covariance matrix was obtained from the same numerical simulation used as the reference (ROOI

with IBI), and in two suboptimal configurations: one in which the covariances were obtained from a high resolution numerical simulation (ROOI with GLORYS-HR), which is supposed to capture the same range of processes than IBI although not exactly, and another one from a low resolution numerical simulation (ROOI with GLORYS-LR) which differs from the reference in the numerical code and also in the resolvable spatial scales.

For the observations and for the reference fields, the native grid of IBI (and GLORYS-HR) was used (1/12 °) (Fig. 2). For the

covariance matrices, the native grid was used for the ROOI with IBI and GLORYS-HR, while for the ROOI with GLORYS-LR data were linearly interpolated to the IBI grid points. The vertical resolution was adapted to a realistic case, emulating ADCP measurements with data every 8 meters. The current vertical profiles were set from -12 m to -148 m and the surface HFR currents at -0.5 m, therefore, the numerical simulation fields were linearly interpolated to the mentioned vertical levels (i.e. -0.5 m, -12 m, -20 m, -28 m, …, -148 m).

### 2.4 Skill assessment

The skills of the data-reconstruction methods were assessed by means of the root mean square difference (RMSD) between the reconstructed fields ($x$) and the reference fields ($y$). The RMSDs were computed at each point of the 3D grid for each study period and for U and V. Thus, for one grid point and a $N$ timesteps ($t$) period:

$$RMSD = \sqrt{\frac{\sum_{t=1}^{N}(x_t - y_t)^2}{N}},$$
(7)

where $x_t$ and $y_t$ are the reconstructed and reference fields at each timestep, respectively.



The relative RMSD to the root mean square (RMS) current (hereinafter RRMSD) was also considered, since the strength and variability of the current are different at different locations of the study area, and therefore, influence the magnitude of the RMSDs. Therefore, the considered relative value is:

$$RRMSD = \frac{RMSD}{RMS} , \qquad (8)$$

where

$$RMS = \sqrt{\frac{\sum_{t=1}^{N}(y_t)^2}{N}} . \qquad (9)$$

Since RMSD and RRMSDs were computed for each study period and for each velocity component, hereinafter we use RMSD-U and RRMSD-U as RMSD and RRMSD computed for U and RMSD-V and RRMSD-V as RMSD and RRMSD computed for V. When the RRMSD is equal to 1 at one point for a study period, it means that the RMSD equals the RMS of the studied period at that point.

## 3 Results and discussion

### 3.1 Mapping the spatio-temporal variability in the study case

In this section, the characteristics of the emulated currents in the study area are analysed in terms of spatial correlation length scales and temporal cross-correlations. The main aim is to provide an overview of the emulated currents used to test the 3D data-reconstruction methods as ground information, in order to justify the scenarios and to support the discussion on their performances. Indeed, the best performances are expected in the areas and periods of higher cross-correlation between different locations and vertical levels.

The correlation ($R$) between two variables $x_1$ and $x_2$ is defined as follows:

$$R(x_1, x_2) = \frac{E[(x_1 - \mu_1) \cdot (x_2 - \mu_2)]}{\sqrt{E[(x_1 - \mu_1) \cdot (x_1 - \mu_1)] \cdot E[(x_2 - \mu_2) \cdot (x_2 - \mu_2)]}} , \qquad (10)$$

where $\mu_i$ is the mean value of $x_i$, that is, $\mu_i = E[x_i]$. In this study, the correlation was used to estimate the relationships between the emulated horizontal currents in two different ways: by means of spatial relationships, determined by spatial correlation length scales (horizontal and vertical), and by means of temporal relationships, determined by temporal cross-correlations between two different points for a certain period of time. Note that for all the correlations presented here the confidence level considered is 95 %.

The spatial correlation length scales are the maximum distances between the grid points where the currents can be considered that are related. These scales were calculated for each velocity component, considering meridional and zonal directions for the computation by means of the e-folding method (described in Ha et al., 2007). If we consider one grid, one velocity component and one direction for the computation we can obtain one $R$ value for each fixed distance between the grid points. That is, if for example we consider the zonal direction and the U component, $x_1$ will be the value of U at each grid point and $x_2$ will be the value of U at the grid point that is at a fixed distance away (a certain number of grid points in the zonal direction) from the





grid point where $x_1$ is evaluated. Therefore, we will obtain one $R$ value for a fixed distance. Then, $R$ is estimated for all the possible distances, thus obtaining correlation values depending on the distance between the grid points. This operation can be repeated for different time steps through a time period, obtaining a correlation vs distance profile for each time step. All these profiles are then averaged for the time period that interests us, obtaining an averaged correlation vs distance profile. In order

to determine the spatial correlation length scale, as explained in Ha et al. (2007), a cut-off point is assumed in the averaged profile where the correlation coefficient decrease to $e^{-1}$ times its original value.

In the water column, U presents higher vertical spatial correlation length scales than V for both buoys (see Table 3) since both are in the Spanish slope, where the zonal slope current dominates the circulation in the area. In addition, these scales are even higher for Matxitxako, which is under a stronger influence of the slope current (with high along-slope spatial correlation)

(Rubio et al., 2013; Solabarrieta et al., 2014). For both velocity components, the scales are larger in winter than in summer when the water column is well-mixed and stratified, respectively (Rubio et al., 2013; Rubio et al., 2019).

With regard to surface currents, the horizontal spatial correlation length scales are higher when the direction of the velocity component and the considered direction for the computation of the correlation (zonal or meridional) are the same, being the highest for U (see Table 3). This is due to the slope current that flows in zonal direction along the Spanish coast (W-E coast)

and in meridional direction along the French coast (N-S coast). Moreover, the scales are slightly larger in winter than in summer, when the slope current is stronger and more persistent (Solabarrieta et al., 2014).

Regarding the temporal relationships, the temporal cross-correlation is defined as the correlation of a variable between two different points of a grid for a period of time, that is, the correlation value $R$ between a variable at one point ($x_1$) and a variable at another point ($x_2$) throughout the period of time that interests us.

For the temporal cross-correlation profiles between the surface and the water column levels at the buoy points, U (Fig. 3a and b) shows higher correlations along the water column in Matxitxako than in Donostia, which agrees with a stronger influence of the slope current at Matxitxako location. For V (Fig. 3c and d) higher correlations are observed in Donostia than in Matxitxako probably due to the stronger signal of this velocity component in that area where there is a change of direction on the bathymetry from zonal direction along the Spanish slope to meridional direction along the French slope. As expected, the

correlation decreases with depth and it is seasonally modulated in coherence with the water column properties in the area (higher stratification decouples surface from sub-surface processes).

The temporal cross-correlation maps between the ADCP vertical levels and the surface points of the HFR grid for U (Fig. 4) show that there is a high temporal correlation between Matxitxako and Donostia velocities and in their nearby areas over the Spanish shelf and slope along the whole analysed water column. For both sites it seems that the core of the slope current is

well sampled with a stronger signature of such current in Matxitxako. With regard to the seasonality, the correlation is higher and more extended to higher latitudes in winter than in summer, when, as mentioned before, the slope current is stronger and more persistent.

For V there is a low (almost null) correlation between Matxitxako and Donostia locations (Fig. 5) and along the Spanish shelf and slope. Higher correlations are found in areas closer to the buoys, which extend to the north in the meridional direction. For



Matxitxako, this fact is more remarkable in summer than in winter, when the current is not so constrained to the Spanish slope. For Donostia, correlations are high along the French shelf and slope, especially in winter, showing how the slope current follows the bathymetry. The maps show that for both velocity components, the decrease of the correlation in depth is stronger in summer than in winter, showing the stratified and the mixed water column periods, respectively.

It is worth highlighting that the emulated spatial correlation length scales and temporal cross-correlations are coherent with the ones obtained using real observations (Rubio et al., 2019; see also supplementary material S1), thus validating further the use of IBI to emulate the study case of the SE-BoB observatory.

**3.2 Data-reconstruction**

Considering all the analysed depths along the water column (from surface to -150 m), both study periods and both data-
reconstruction methods, satisfactory reconstructions are obtained. These reconstructions provide mean RMSDs for each depth (Figs. 10-11) ranging from 0.55 (0.7) cm s$^{-1}$ to 10.94 (9.58) cm s$^{-1}$ for the whole (reduced) grid and mean RRMSDs ranging from 0.07 (0.12) to 3.47 (1.31) with typical values around 1 or less, that is, with reconstructed field errors around the RMS or less at each point. Although, in general, the RRMDs are increased with depth (thus, showing a worse performance), mean RMSDs lower than 10.94 cm s$^{-1}$ are obtained at -150 m.

The results obtained in this study are summarized in Table 4, where the spatial mean RMSDs and RRMSDs are shown for three different depths, for both study periods, for both data-reconstruction methods (the ROOI with GLORYS-LR) and for the whole and reduced grids.

It is observed that, in general, the RMSDs and the RRMSDs are affected by the spatial and temporal variability of the current regime for the study area described in Sect. 3.1. The mean RMSDs are higher in winter than in summer because currents are
more intense in that period, however, the RMSs are also higher in that period and in relative terms the reconstructions show better results in winter (lower mean RRMSDs). This dependence of the results on the current regime can be also observed if we compare the reduced and the whole grid cases. For the reduced grid case, that covers an area of intense zonal slope currents, highest mean RMSDs and lowest mean RRMSDs are obtained for U. Since V is much weaker for this grid, it provides the lowest mean RMSDs, nevertheless, the expected increase in the mean RRMSDs is not so clear compared to the whole grid
case due to lower RMSs.

Regarding the comparison between both methods, for the whole grid case, the mean RRMSD-Us are remarkably higher for the DCT-PLS, whereas the RRMSD-V provides similar results for both. On the other hand, for the reduced grid case the results for RRMSD-U for the DCT-PLS are better, showing that this method performs better in well sampled areas while the ROOI performs better than the DCT-PLS in the rest of the areas (although it also performs well in the former area).

All these results, in addition to more specific analyses, are shown below in terms of RRMSDs by means of maps (Figs. 6-9) and horizontal mean values' graphs along the water column (Figs. 10-11). The results of the RMSDs are shown in supplementary material S4. For the ROOI RRMSD maps, the results with the spatial covariances from GLORYS-LR are



presented because those are the ones that most challenge the method. In fact, for the ROOI with GLORYS-HR the RRMSDs are even lower (see supplementary material S2), however the main conclusions are very similar.

For the ROOI, the RRMSD spatial distribution is more uniform in summer (Fig. 6) than in winter (Fig. 7) due to the more variable summer current regime. The Spanish slope area shows the lowest RRMSD-Us due to the strong signal of the along-

slope current, with lower values in winter than in summer, suggesting that the reconstructed fields are more accurate in well sampled areas and that U is well resolved in the numerical simulations used for the definition of the spatial covariances. For the RRMSD-V, the French slope and part of its platform show the lowest values in winter, indicating that the slope current is well reconstructed for that period. Since the density of the observation is much higher at the surface, it is expected the method to perform better in the upper layers, in fact, it is observed that the RRMSDs increase with depth. This increase is more

remarkable for summer than for winter, probably due to higher vertical shear in the currents due to the stratification conditions. It is shown that for the ROOI with GLORYS-LR, the RRMSDs are below 1.25, that is, the RMSD is below 1.25 times the RMS at each point, except for some concrete areas.

With regard to the DCT-PLS RRMSD maps (Figs. 8-9), it is observed that the values are the lowest near the surface and the buoys locations, showing that this method's skills are better adjusted in well sampled areas. For both velocity components, the

RRMSDs are lower in winter (Fig. 9), being characterized by stronger and persistent currents compared to the summer regime (Fig. 8). For the RRMSD-U, the Spanish slope area shows the lowest values for both periods, whereas low RRMSD-Vs are observed over the French slope in winter, showing that this method is also able to reconstruct the slope current. Overall, RRMSDs increase with depth, nevertheless, for RRMSD-V in summer, the values are higher for -52 m (Fig. 8d) than for -100 m (Fig. 8f). This could be related to the water depths where the vertical shear of the currents is expected to be the highest due

to the presence of the seasonal thermocline, which in this period is located between -30 m and -50 m. For the DCT-PLS, the RRMSDs are not as smooth as for the ROOI, with RMSDs near the observation areas lower than half the RMS at each point and values out of those areas higher than twice the RMS at each point.

Thus, for both methods lower RRMSDs are observed in winter than in summer and along the slope for the along-slope component of the velocity; with RRMSDs increasing with depth. While, the DCT-PLS is more effective at well sampled areas,

the ROOI performs better for the rest of the areas. In general, the best performances are located in the areas where correlations between velocities in the ADCP locations and the rest of the grid locations are high (Figs. 3-5), showing that this a-priori analysis (shown in Sect. 3.1) can provide an approximate idea about the areas where the data-reconstructions could, in principle, perform better.

It is observed that the results for the DCT-PLS worsen quickly as we get away from the observation points. Considering the -

52 m depth layer, we observe that RRMSD values obtained with the DCT-PLS method increase to 0.25 at ~31 km (6.3 km) for the U(V) component in the zonal (meridional) direction.

The analysis of the spatial mean of RRMSDs with depth (Figs. 10-11) aimed to compare both data-reconstruction methods' skills, regardless of the spatial variability shown in previous figures. Note that the ROOI with both IBI and GLORYS are



shown in this analysis, and that the same grid points were considered for all the methods and products. The whole grid and the reduced grid (see Fig. 1b) were considered in order to explore the sensitivity of the results to the choice of different areas.

For the whole grid case (Fig. 10), the ROOI with IBI and GLORYS-HR performs better for both velocity components, whereas, the ROOI with GLORYS-LR provides similar results as the DCT-PLS for V (Fig. 10b and d) and it provides much better

results for U (Fig. 10a and c). In addition, as it could be noticed in Table 4 and along Figs. 6-9, the mean RRMSDs show RMSDs around 1 or less times the RMS at each point, except for U for the DCT-PLS.

In the reduced grid case (Fig. 11), the lowest mean RRMSD-Us are observed for the DCT-PLS, working significantly better than the ROOI. The mean RMSDs are around 0.75 or less times the RMS at each point, with values around 0.5 or less for the DCT-PLS, providing quite a satisfactory reconstruction of the along-slope velocity component in the Spanish slope area. Thus,

if we consider the whole water column, the ROOI provides again smaller RRMSDs than the DCT-PLS for the whole grid case, whereas, the DCT-PLS provides better results in well sampled areas.

With regard to the seasonal analysis, as observed in the maps (Figs. 6-9), overall, for both data-reconstruction methods and for both velocity components, Figs. 10-11 show lower RRMSDs in winter than in summer, except for the RRMSD-U in the whole grid case for the DCT-PLS (Figs. 10 a and c). This exception was caused by the big RRMSDs over the French shelf and slope

for that period (see supplementary material S3), since this method expands the zonal component to that area of meridional regime.

### 3.3 Sensitivity test: increased number of ADCPs

An analysis with observations of ADCPs in two additional locations was carried out in order evaluate the sensitivity of the data-reconstruction methods to an increased number of observations with higher geographical coverage. The two extra ADCPs

were located over the French slope, since this could be a strategic area to monitor the winter slope current downstream the Capbreton canyon.

Only the winter period is shown, when the slope current is the strongest and the effects of the new scenario are more noticeable. We selected here only the results obtained for the -52 m layer, due to its representativeness of the changes between the 2-buoy and the 4-buoy scenario for all the water column levels analysed before. The performance of the methods for this configuration

is shown subtracting the RRMSD maps of the 2-buoy case to the 4-buoy case. Therefore, the negative (positive) values in Fig. 12 show that the RRMSD is lower (higher) for the 4-buoy configuration, thus, showing a better (worse) performance. In general, this new scenario improves the performance of both data-reconstruction methods with smoother changes for the ROOI, since it already uses historical information of the covariances in the whole study area, including the locations of the extra ADCPs.

For U, the addition of two extra ADPC profiles does not affect the Spanish slope area where there are already two moorings that capture the slope current. In the rest of the grid, for the DCT-PLS, the performance of the reconstruction is remarkably improved; whereas, for the ROOI, although in general the reconstruction is improved, there are some specific areas where the RRMSD-Us are slightly increased.





For V, the results improve along the French slope, which are more remarkable for the DCT-PLS. However, for this method, the RRMSD-Vs are increased in the areas close to that slope, probably due to the spread signal of the slope observations to those nearby areas which are not affected by the slope current regime.

All in all, the reconstructions of the along-slope component are improved with the additional observations.

## 4 Summary and Conclusions

In this paper we investigated the combination of data obtained from multiplatform observing systems to reconstruct the 3D velocity field in a shelf-slope region by means of two data-reconstruction methods. More precisely, we investigated the case of combining surface velocities obtained from a HFR system and velocity profiles measured by two moorings equipped with ADCPs, which is a typical configuration among the existing coastal observatories. The performances of the reconstructions were assessed through a controlled reality experiment in which observations were extracted from a numerical simulation and the results compared to the original simulated fields.

A preliminary spatial correlation length scale analysis and a temporal cross-correlation analysis provided a first guess of the areas where the reconstructions could perform better. We obtained satisfactory reconstruction results with spatial mean RMSDs ranging between 0.55–10.94 cm s$^{-1}$, for the first 150 m depth, which represents a relative error of 0.07–3.47 times the RMS current at each point. In addition, the results show that the main feature of the region, the slope current, is well reconstructed by both methods. The results have also shown that the addition of two additional moorings can improve significantly the results of the reconstruction.

Regarding the data-reconstruction methods, each one has its pros and cons. The DCT-PLS is only fed with the observations with no extra information about the study area, so its configuration is simpler. It performs well in well sampled areas, but its quality is quickly degraded elsewhere. On the other hand, the ROOI is a robust data-reconstruction method that uses historical information of the study area, and thus provides better results in areas which are not sampled. The shortcoming of this method is that it needs historical information of the study area. This is typically obtained from a realistic numerical simulation of the region although it does not need to be contemporary to the observational period. Also, the method requires more tuning, so its implementation demands a careful testing of the parameters.

These data-reconstruction methods have proven to be reliable and could be used in a wide range of applications. For instance, due to their low computational cost they could be used to obtain new operational products, combining data from different sources and complementary spatial coverage in near real time. Moreover, through Observing System Simulation Experiments and Observing System Experiments (OSSE and OSE) an optimization of existing observing networks can be proposed, providing a potential tool for taking decisions for future planning of coastal observatories or to set-up optimal operational data assimilation strategies. Additionally, the 3D velocity reconstructions might have applications for coastal risk assessment or for model validation, as well as for broadening the utility of coastal observing systems to biological, geochemical and environmental issues.
**Data availability**

The IBI_REANALYSIS_PHYS_005_002 product is available on the CMEMS website (http://marine.copernicus.eu/services-portfolio/access-to-products/?option=com_csw&view=details&product_id=IBI_REANALYSIS_PHYS_005_002).

The GLOBAL_REANALYSIS_PHY_001_025 product is available on the CMEMS website (http://marine.copernicus.eu/services-portfolio/access-to-products/?option=com_csw&view=details&product_id=GLOBAL_REANALYSIS_PHY_001_025).

The GLOBAL_REANALYSIS_PHY_001_030 product is available on the CMEMS website (http://marine.copernicus.eu/services-portfolio/access-to-products/?option=com_csw&view=details&product_id=GLOBAL_REANALYSIS_PHY_001_030).

**Author Contributions**

IMN, EF, GJ, MB, AG, AC, AR: contribution to the main structure and contents. In addition, IMN produced the figures and EF, GJ provided the software, the tools and gave advice for the reconstruction with the DCT-PLS and ROOI methods, respectively.

**Competing interests**

The authors declare that they have no conflict of interest.

**Acknowledgments**

This study has been supported by the JERICO-NEXT project, funded by the European Union's Horizon 2020 research and innovation programme under grant agreement No 654410. This study has been also undertaken with the financial support of the Department of Environment, Regional Planning, Agriculture and Fisheries of the Basque Government (Marco Program). I. Manso was supported by a PhD fellowship from also the Department of Environment, Regional Planning, Agriculture and Fisheries of the Basque Government. This study has been conducted using E.U. Copernicus Marine Service Information. This is contribution number XXX, of the Marine Research Division of AZTI-Tecnalia.

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



**Table 1.** List of some data gap-filling/reconstruction methods applied to oceanographic data sets.

| Name of the method | Reference |
| --- | --- |
| Open-boundary Modal Analysis (OMA) | Kaplan and Lekien, 2007 |
| Data Interpolating Empirical Orthogonal Functions (DINEOF) | Beckers and Rixen, 2003 and Alvera-Azcárate et al., 2005 |
| Self-Organizing Maps (SOM) | Kohonen, 1982, 1997 |
| Variational Analysis (VA) | Sasaki, 1970 and Wahba and Wendelberger, 1980 |
| Optimal Interpolation (OI) | Gandin, 1965 |
| Discrete Cosine Transform Penalized Least Square (DCT-PLS) | García, 2010 |
| Reduced Order Optimal Interpolation (ROOI) | Kaplan et al., 1997 |





**Table 2.** Details of the three numerical simulations used in this study.

| | IBI | GLORYS-LR | GLORYS-HR |
|---|---|---|---|
| **Product identifier** | IBI_REANALYSIS_PHYS_005_002 | GLOBAL_REANALYSIS_PHY_001_025 | GLOBAL_REANALYSIS_PHY_001_030 |
| **Regional / Global** | Regional | Global | Global |
| **Spatial resolution** | 0.083º x 0.083º | 0.25º x 0.25º | 0.083º x 0.083º |
| **Temporal resolution** | Daily | Daily | Daily |
| **Model** | NEMO v3.6 | NEMO v3.1 | NEMO v3.1 |
| **Data assimilation** | In-Situ TS Profiles<br>Sea Level<br>SST | Sea Ice Concentration and/or Thickness<br>In-Situ TS Profiles<br>Sea Level<br>SST | Sea Ice Concentration and/or Thickness<br>In-Situ TS Profiles<br>Sea Level<br>SST |
| **Atmospheric forcing** | ECMWF ERA-interim | ECMWF ERA-interim | ECMWF ERA-interim |
| **Bathymetry** | GEBCO_08 + different local Databases | ETOPO1 for deep ocean and GEBCO8 on coast and continental shelf | ETOPO1 for deep ocean and GEBCO8 on coast and continental shelf |
| **Initial conditions** | January 1992: T, S, velocity components and sea surface height from GLORYS2V4 | December 1991: T, S regressed from EN4 | December 1991: T, S regressed from EN.4.2.0 |
| **Open boundary data** | Data from daily outputs from the CMEMS GLOBAL reanalysis eddy resolving system. | … | … |
| **Application in this study** | Observations, reference fields, the covariance matrix for the ROOI | The covariance matrix for the ROOI | The covariance matrix for the ROOI |
| **For a more detailed description:** | http://cmems-resources.cls.fr/documents/PUM/CMEMS-IBI-PUM-005-002.pdf<br><br>http://resources.marine.copernicus.eu/documents/QUID/CMEMS-IBI-QUID-005-002.pdf | http://cmems-resources.cls.fr/documents/PUM/CMEMS-GLO-PUM-001-025.pdf<br><br>http://resources.marine.copernicus.eu/documents/QUID/CMEMS-GLO-QUID-001-025.pdf | http://cmems-resources.cls.fr/documents/PUM/CMEMS-GLO-PUM-001-030.pdf<br><br>http://resources.marine.copernicus.eu/documents/QUID/CMEMS-GLO-QUID-001-030.pdf |

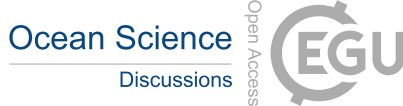

**Table 3.** Seasonal spatial correlation length scales for the emulated current velocity components U and V in the study area, for the summer and winter periods and in zonal and meridional directions. Note that the surface horizontal scales are shown in
15 kilometres and that the vertical scales in depth at Matxitxako and Donostia buoy points are shown in meters.

| Current component | Surface (km) | | | | Depth (m) | | | |
| --- | --- | --- | --- | --- | --- | --- | --- | --- |
| | Summer | | Winter | | Summer | | Winter | |
| | Zonal direction | Meridional direction | Zonal direction | Meridional direction | Matxitxako buoy | Donostia buoy | Matxitxako buoy | Donostia buoy |
| U | 78 | 15 | 79 | 16 | 24 | 23 | 88 | 43 |
| V | 11 | 60 | 12 | 73 | 19 | 15 | 30 | 36 |





**Table 4.** Summary of the results of the reconstructions with ROOI (with GLORYS-LR) and DCT-PLS in terms of spatial mean RMSDs and RRMSDs for the whole and reduced grids, for the summer and winter study periods and for different depths.

| Parameter | Considered grid | | ROOI | | DCT-PLS | |
|---|---|---|---|---|---|---|
| | | | Summer | Winter | Summer | Winter |
| <RMSD> (cm s$^{-1}$) | Whole | U/V -12 m | 3.79/5.08 | 4.46/6.28 | 3.59/3.62 | 3.10/2.65 |
| | | U/V -52 m | 2.84/3.66 | 4.05/5.45 | 4.01/4.48 | 5.69/4.99 |
| | | U/V -100 m | 2.69/3.14 | 3.89/5.31 | 4.10/3.22 | 8.45/5.32 |
| | Reduced | U/V -12 m | 6.35/3.87 | 8.29/3.91 | 4.15/2.77 | 3.92/1.93 |
| | | U/V -52 m | 4.98/2.02 | 9.19/2.85 | 3.10/2.01 | 4.66/2.67 |
| | | U/V -100 m | 4.31/1.77 | 8.38/2.46 | 2.33/1.75 | 3.66/2.59 |
| <RRMSD> | Whole | U/V -12 m | 0.83/0.83 | 0.84/0.88 | 0.88/0.64 | 0.67/0.38 |
| | | U/V -52 m | 0.98/1.02 | 0.94/0.80 | 1.69/1.33 | 1.83/0.74 |
| | | U/V -100 m | 1.05/1.04 | 0.92/0.80 | 1.82/1.07 | 2.79/0.83 |
| | Reduced | U/V -12 m | 0.56/0.94 | 0.53/1.04 | 0.37/0.74 | 0.25/0.53 |
| | | U/V -52 m | 0.79/0.94 | 0.64/0.88 | 0.54/1.03 | 0.33/0.90 |
| | | U/V -100 m | 0.95/1.04 | 0.72/0.80 | 0.54/1.00 | 0.32/0.95 |



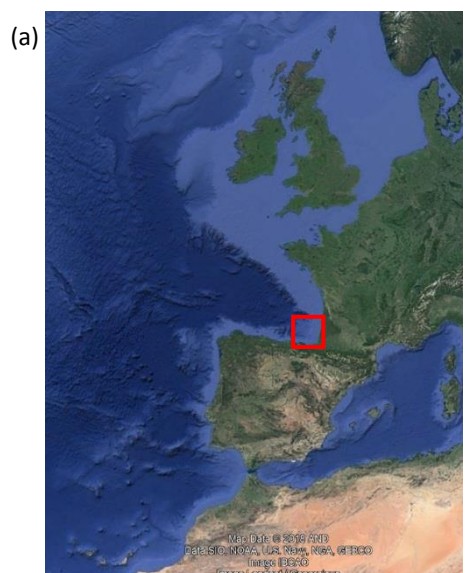

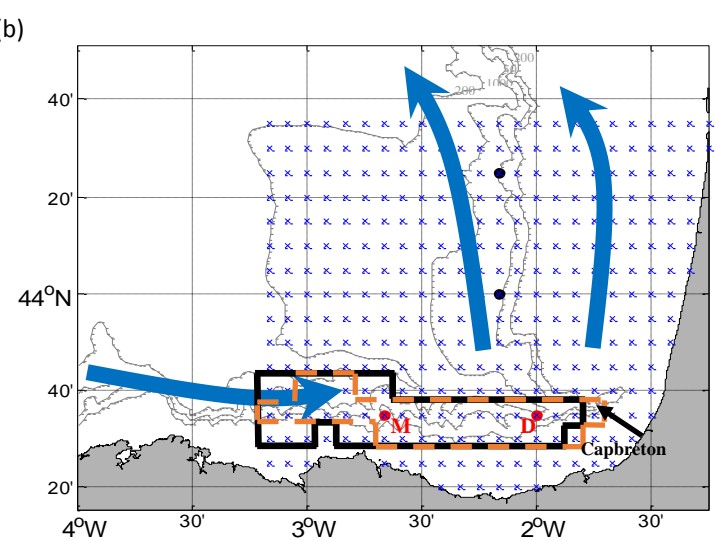

**Figure 1.** (a) Location of the study area (red square). (b) Close-up map of the study area. The winter IPC is represented by blue solid arrows. The grid used for the emulated HFR surface current fields is shown by blue crosses. The red dots provide

10 the location of the current vertical profiles that emulate the EuskOOS buoys: Matxitxako (red M) and Donostia (red D), whereas the black dots depict the location of the two extra buoys used for the 4-buoy scenario. The bold black lines delimit the winter reduced grid, whereas the dashed orange lines delimit the summer one. The grey lines show the 200, 500, 1000 and 2000 m isobaths.





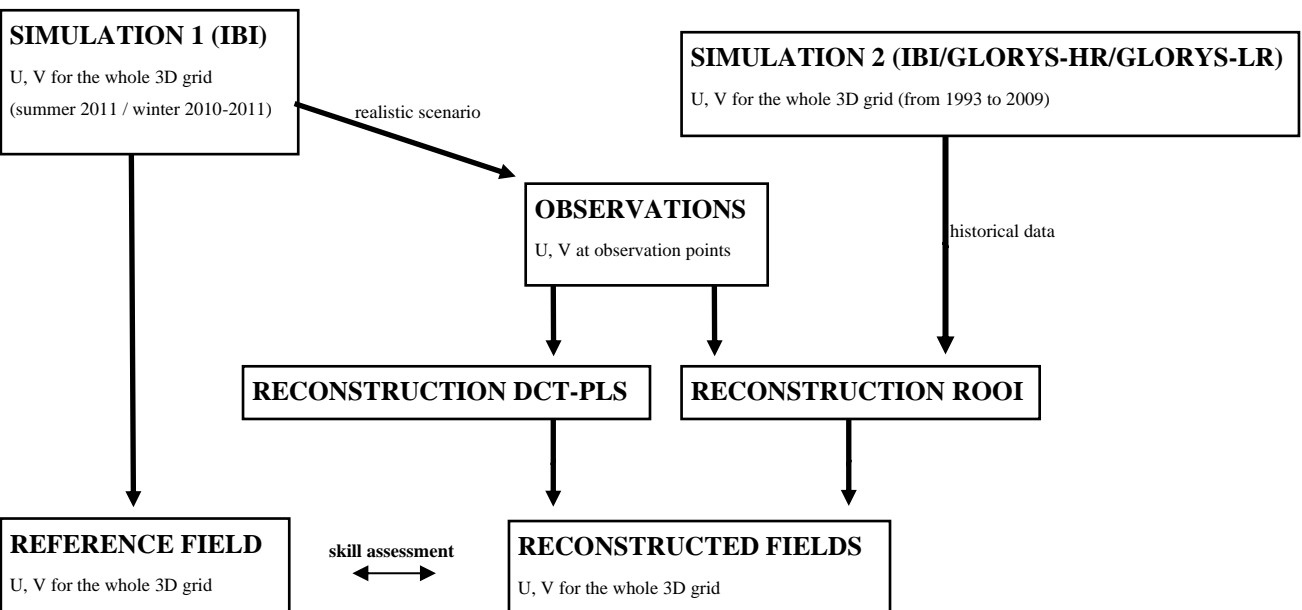

**Figure 2.** Scheme of the approach used to test the performance of the two data-reconstruction methods, that are described in Sect. 2.2. SIMULATION 1 and SIMULATION 2 are presented in Sect. 2.3.





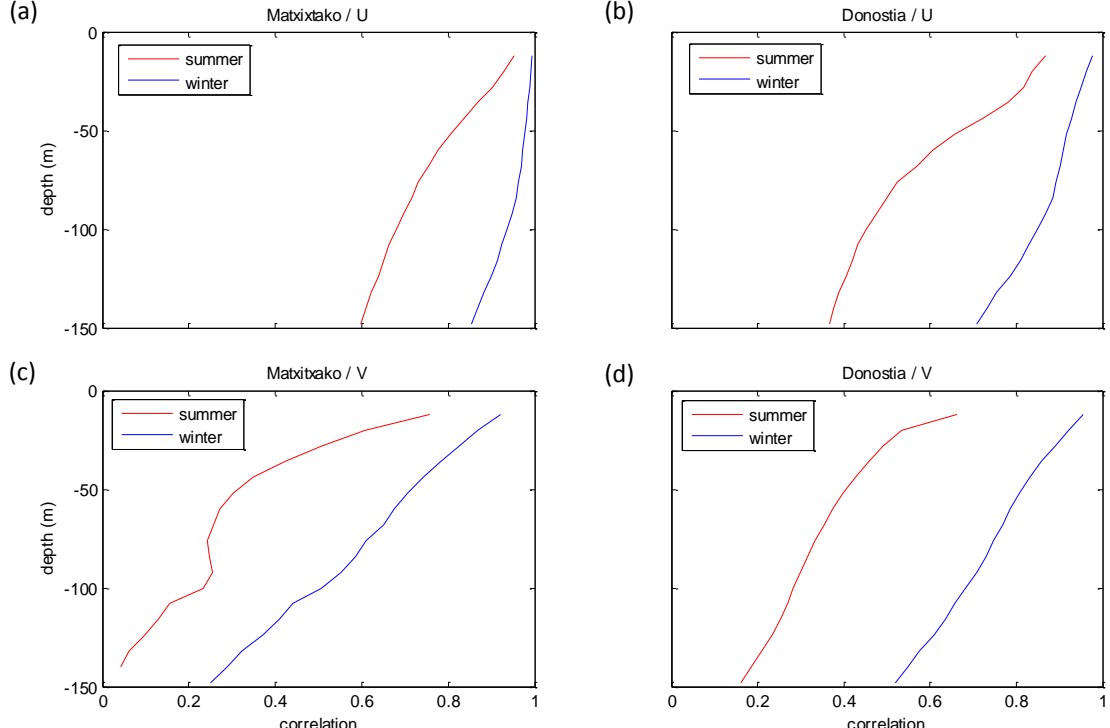

**Figure 3.** U (a, b) and V (c, d) temporal cross-correlation between the surface and the water column levels, for winter (blue)
10 and summer (red) periods. In Matxitxako location (a, c) and in Donostia location (b, d).


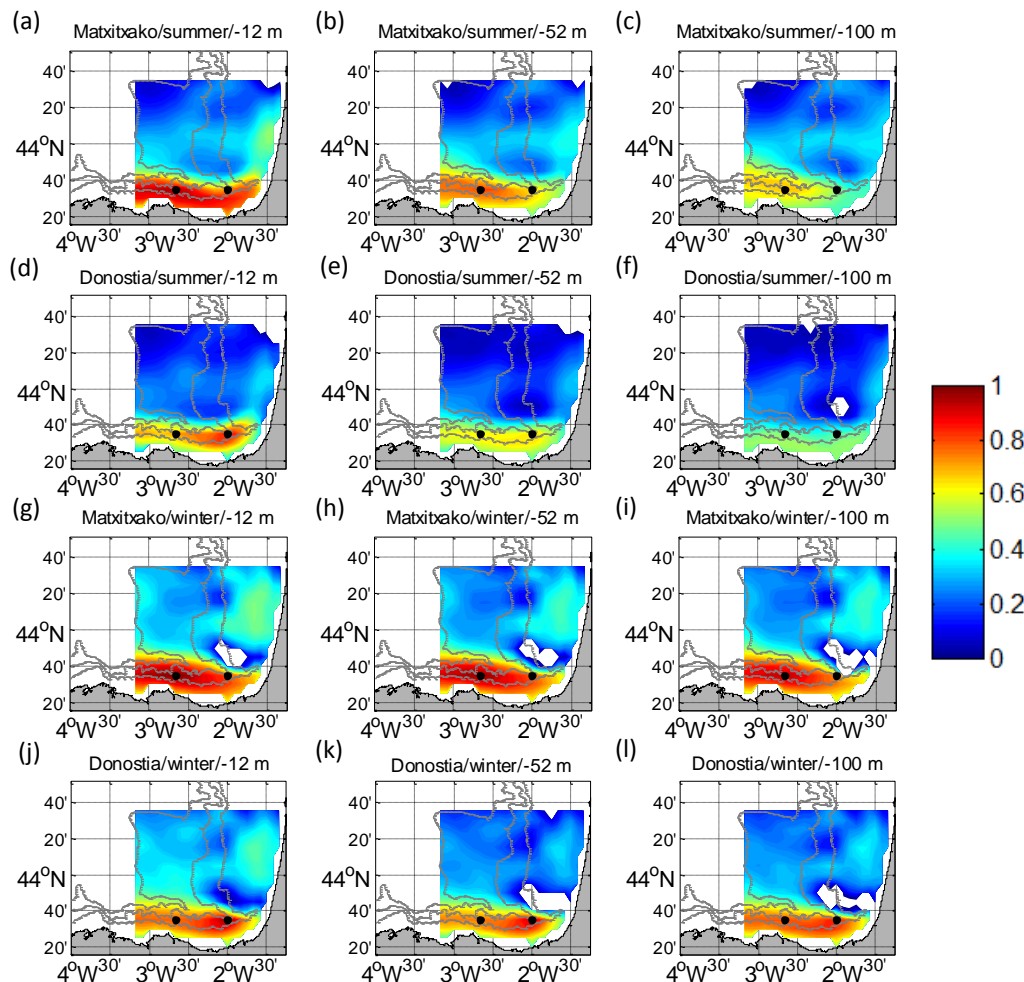

**Figure 4.** Temporal cross-correlation maps between the ADCP vertical levels and the surface points of the HFR grid for U. a, b, c, g, h, i for Matxitxako buoy and d, e, f, j, k, l for Donostia buoy. Different depths considered: -12 m (a, d, g, j), -52 m (b, e, h, k) and -100 m (c, f, i, l), for summer (a-f) and winter (g-l). The white gaps are the areas where the confidence level is less than 95%. The black dots depict the locations of the ADCPs.





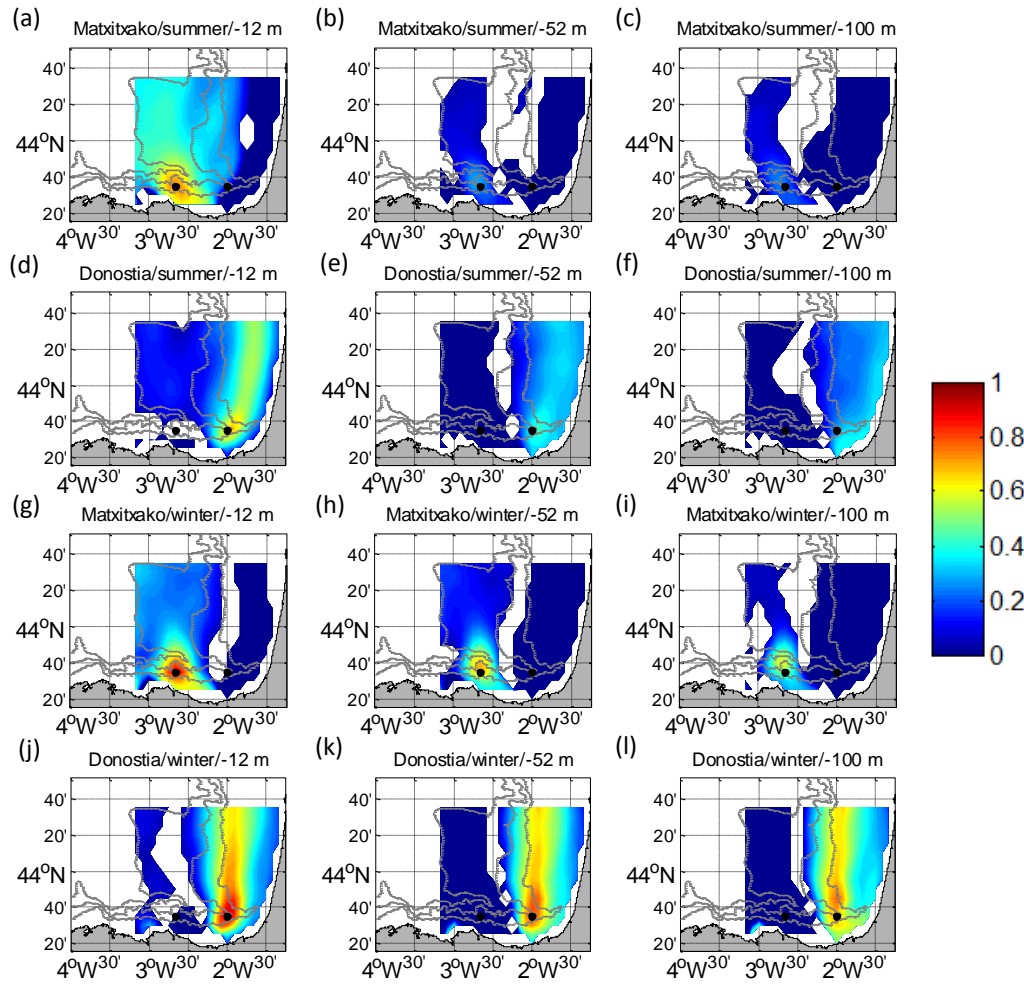

5 **Figure 5.** Temporal cross-correlation maps between the ADCP vertical levels and the surface points of the HFR grid for V. a, b, c, g, h, i for Matxitxako buoy and d, e, f, j, k, l for Donostia buoy. Different depths considered: -12 m (a, d, g, j), -52 m (b, e, h, k) and -100 m (c, f, i, l), for summer (a-f) and winter (g-l). The white gaps are the areas where the confidence level is less than 95%. The black dots depict the locations of the ADCPs.


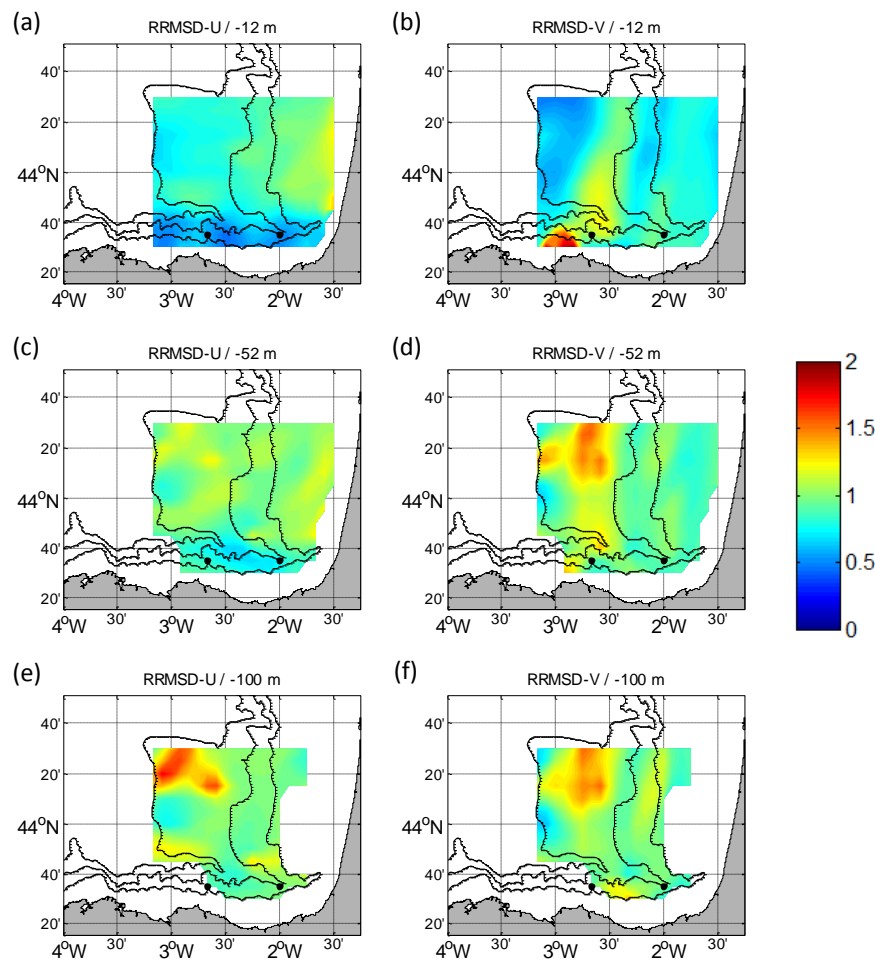

**Figure 6.** RRMSD maps for the summer period between the reference fields and the outputs of the ROOI with GLORYS-LR for U (a, c, e) and V (b, d, f). Different depths considered: -12 m (a, b), -52 m (c, d) and -100 m (e, f). The black dots depict the locations of the ADCPs.


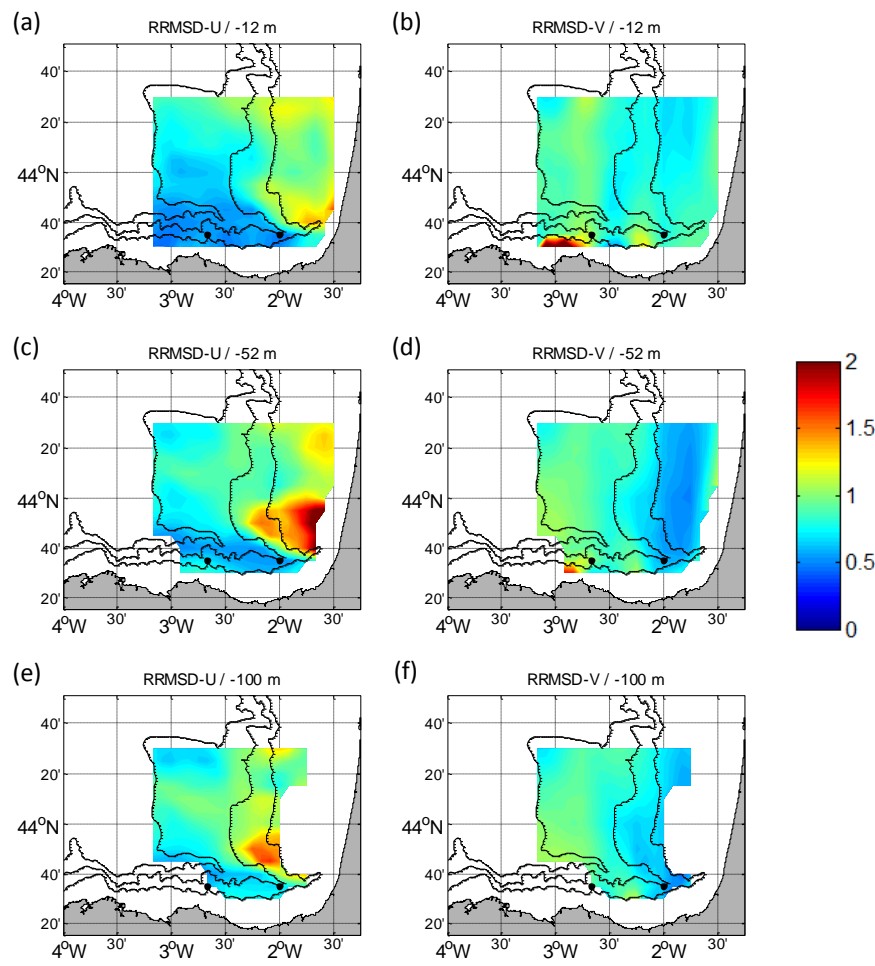

**Figure 7.** RRMSD maps for the winter period between the reference fields and the outputs of the ROOI with GLORYS-LR
for U (a, c, e) and V (b, d, f). Different depths considered: -12 m (a, b), -52 m (c, d) and -100 m (e, f). The black dots depict
the locations of the ADCPs.

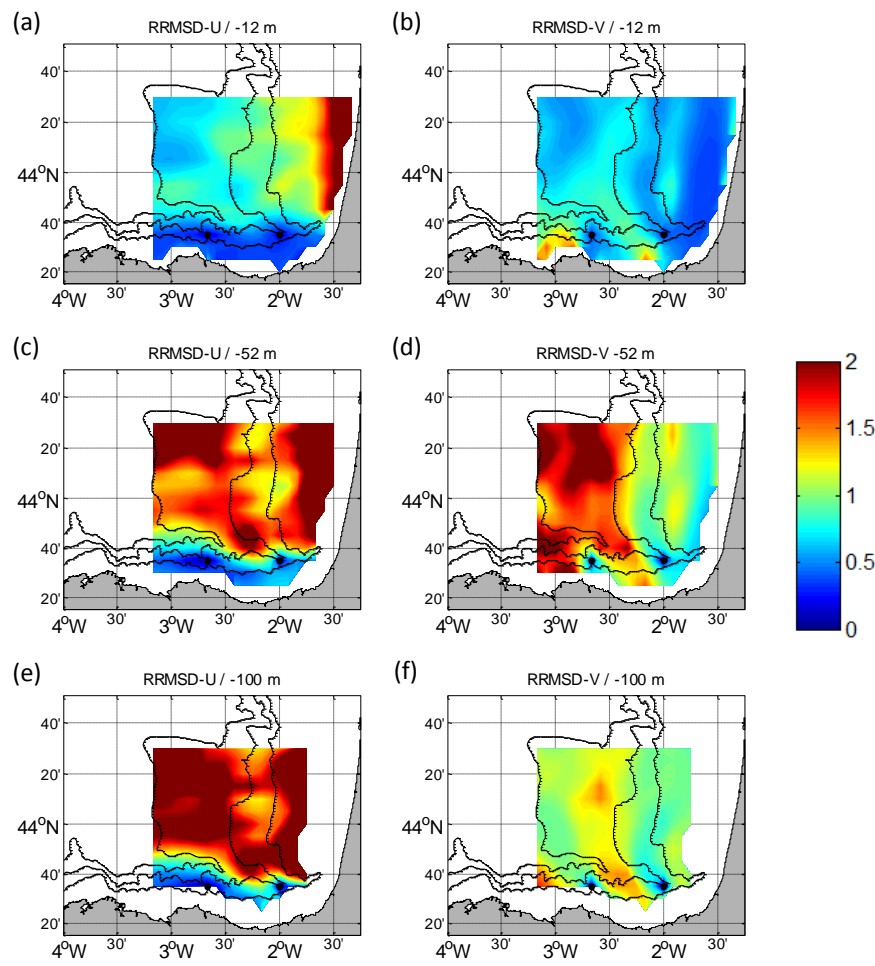

**Figure 8.** RRMSD maps for the summer period between the reference fields and the outputs of the DCT-PLS for U (a, c, e) and V (b, d, f). Different depths considered: -12 m (a, b), -52 m (c, d) and -100 m (e, f). The black dots depict the locations of the ADCPs.





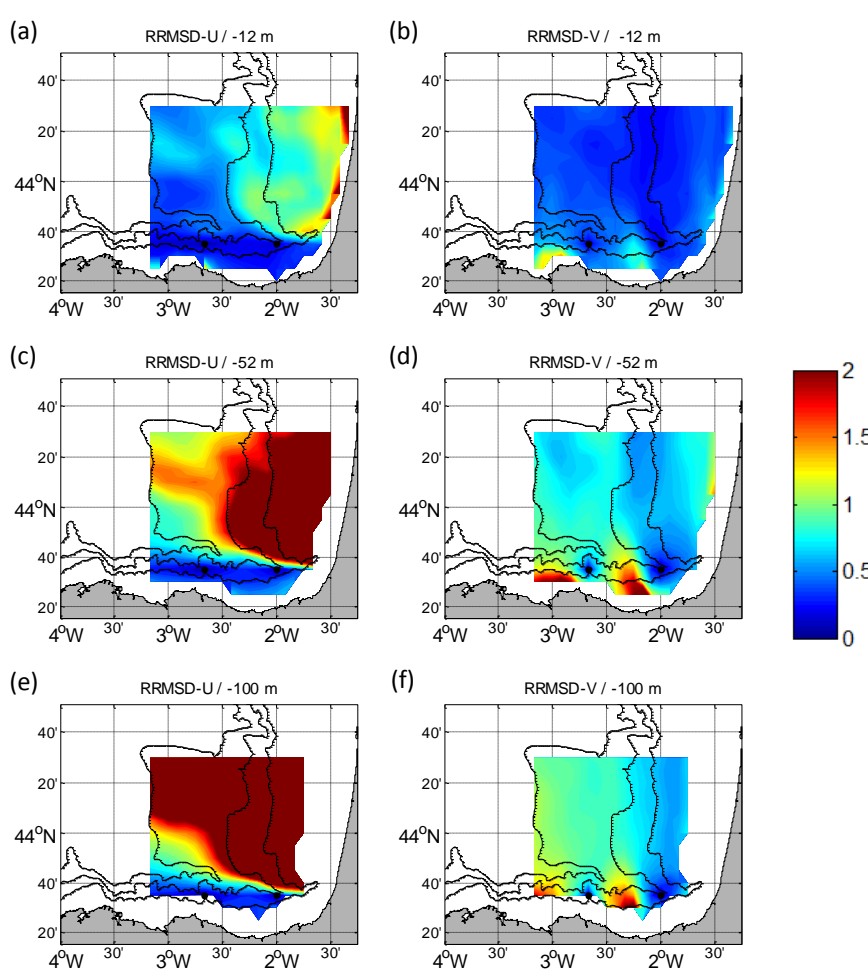

5 **Figure 9.** RRMSD maps for the winter period between the reference fields and the outputs of the DCT-PLS for U (a, c, e) and V (b, d, f). Different depths considered: -12 m (a, b), -52 m (c, d) and -100 m (e, f). The black dots depict the locations of the ADCPs.



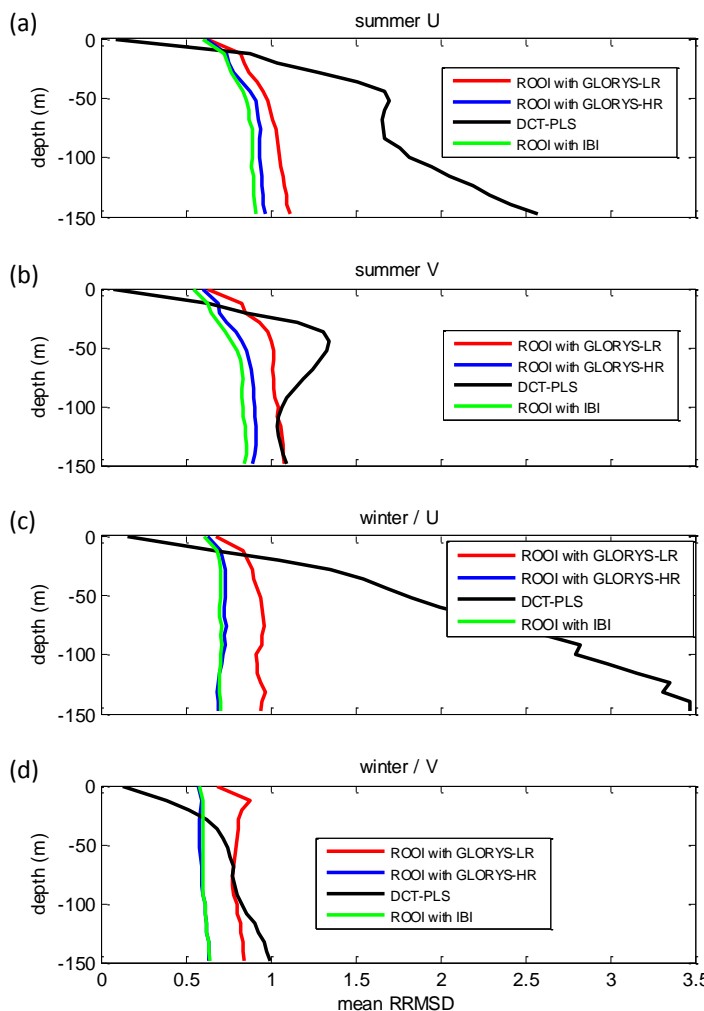

**Figure 10.** Mean RRMSDs related to all the data-reconstruction methods for each depth considering the whole grid. For the summer period (a, b) and for the winter period (c, d). U in a, c and V in b, d.





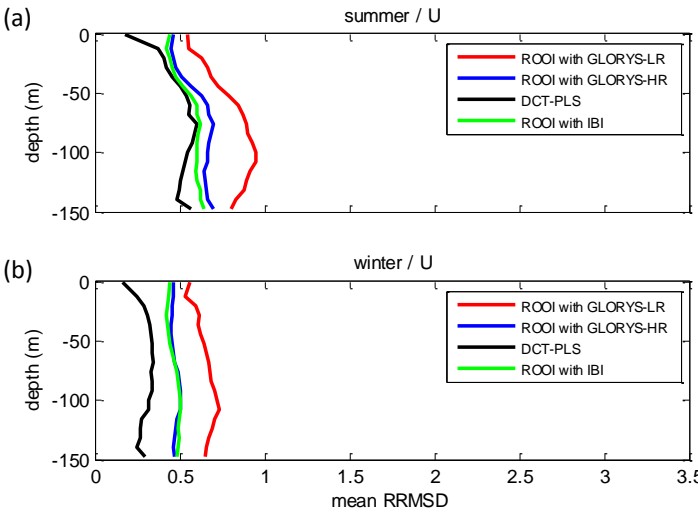

**Figure 11.** Mean RRMSD-U related to all the data-reconstruction methods for each depth considering the reduced grid domain.

10  For the summer period (a) and for the winter period (b).

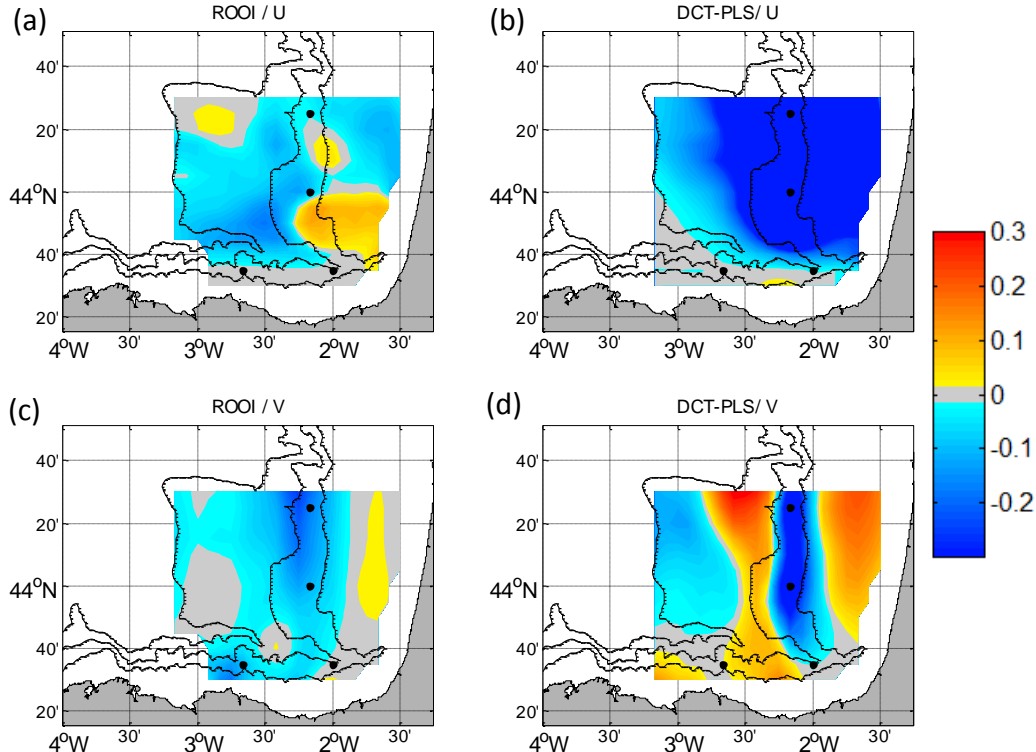

**Figure 12.** The 4-buoy scenario RRMSD maps subtracted by the 2-buoy scenario RRMSD maps for winter at -52 m, therefore, negative values mean better performance with the 4-buoy configuration. For U (a, b) and for V (c, d). For the ROOI (a, c) and for the DCT-PLS (b, d). The black dots depict the locations of the ADCPs.

