# Peer review of "3D Reconstruction of Ocean Velocity from HFR and ADCP: a model-based assessment study"

_Ocean Science, 2019_

## Referee Comment (RC1) · Anonymous Referee #1 · 4 Nov 2019

Review of "Three-Dimensional Reconstruction of Ocean Circulation from Coastal Marine Observations: Challenges and Methods"

Line 7,          should technology be capitalized?

Line 16,         should it read multiplatform or multisensory
                 Change to "aiming for the continuous"

Line 17,         change from "is today" to "are"

Line 18,         change to "resolution, but are limited to the"

The authors should make it more clear that the the ADCP and HFR data they are discussing were derived from model output.  On the first read of the manuscript, I thought that data was from sensors deployed in the ocean.

Line 27,         change to "combining simulated information from "

Line 30,         change to "performance"

Line 17,         change to "with surface temperatures over"

Line 1,          remove Moreover

Line 3,          replace Summarizing with In summary,

Line 11,         change to "surface current fields along the Mid Atlantic"

Line 19,         can the authors be more specific on what is meant by the observations and the reference fields

What data source was used for the correlation scale tests, IBI, GLORYS-HR or GLORYS-LR?

Figs 10-11 are mentioned before Figs 6-9, can this be changed

Line 6 change to "the combination of synthetic data that mimics sensors from a multiplatform observing system to reconstruct"

---

## Referee Comment (RC2) · Anonymous Referee #2 · 26 Nov 2019

I went through the manuscript with great interest as the data-driven reconstruction of subsurface velocity is a topic of grea interest and has some potential but I must admit I was disappointed when reading the body of the text. the title and the paper are misleading as they suggest that real data are used for the task, which unfortunely is not the case here. The manuscript indeed focuses on 'emulated' observations of currents provided through some 'supposedly' accurate and realistic model simulation. However, when it comes to the description of the model, the reader is pointed out to some references to other studies. If you have the model, why not compare that to the HFR data if you do not want to use the data itself for the task?

[Figure]

the literature review is lacking some important references. Development of subsurface current estimation procedures to complement surface currents started as soon as radar technologies were available. Some are given below, I leave the Authors to do a thoroughly review. Simple models dedicated to the prediction of current profiles have been developed (Prandle D., 1982. The vertical structure of tidal currents. Geophysical and Astrophysical Fluid Dynamics, 22, 29-49, 1982. Prandle D., 1987. The fine-structure of nearshore tidal and residual cirrculatins revealed by HF radar surface current measurements. Journal of Physical Oceanography, 17, 231-245, 1987. Prandle D., 1991. A view of near-shore dynamics based on observations from HF radar. Progress in Oceanography, 27, 403-438, 1991. ; Davies, 1982, 1983, 1985, 1992). Semi-empirical models, based on shallow-water hydrodynamics coupled to a modal representation of the current profiles, in which the modes have been estimated from local current profiles time series, have been used to estimate the 3-dimensional flow field from HF surface currents near the Rhine river outflow (de Valk C.F., 1999. Estimation of the 3-D current fields near the Rhine outflow from HF radar surface current data. Coastal Engineering, 37, 487-511, 1999.). A statistical method, based on vector correlation analysis between HF surface and ADCP subsurface currents and coupled with a modal representation in which modes were obtained from ADCP currents, was proposed in order to "project" surface currents along the water column. A different approach that infers the approximate shape of the current profiles from surface data without making use of local current profiles, has been introduced in 2001 for shallow-water coastal zone (Shen et Evans, 2001), subsequently extended to deep-water regions (Shen C.Y., Evans T., 2001. Surface-to-subsurface velocity projection for shallow water currents. Journal of Geophysical Research, 106, C4, 6973-6984, 2001. Shen C.Y., Evans T., 2002. Dynamically constrained projection for subsurface current velocity. Journal of Geophysical Research, 107, C11, 3203-3216, 2002. Doi: 10.1029/2001JC001036.), and is meant as an alternative to data assimilation into circulation models. The same approach has been recently applied to a shallow-water region in order to infer current profiles and to obtain maps of sea-surface slope from HF radar current estimates (Marmorino G.O.,

Shen C.Y., Evans T., Lindemann G.J., Hallock Z.R., Shay L.K., 2004. Use of 'velocity projection' to estimate the variation of sea-surface height from HF Doppler radar current measurements. Continental Shelf Research, 24, 353-374, 2004.). More recently, coupled with a two-layer density plume model, this technique was applied to estimate current profiles and density structure in a coastal zone dominated by a plume (Gangopadhyay A., Shen C.Y., Marmorino G.O., Mied R.P., Lindeman G.J., 2005. An extended velocity projection method for estimating the subsurface current and density structure for coastal plume regions: an application to the Chesapeake Bay outflow). The so-called "Velocity Projection Technique" introduced in these papers, relies on the surface-to- subsurface viscous coupling and turbulent transfer of momentum and shear in order to infer the velocity distribution over depth from measured surface currents and wind stress. This method, applied in its original formulation to shallow coastal water, resolves the vertical structure of the currents in terms of a finite expansion of orthogonal modes spanning the water column. The modal weights are obtained by applying appropriate dynamical constraints to the inferred current profiles and their vertical derivatives at the boundaries.

Abstract / main body: define surface. HFR sense different 'depths' based on the working frequency. Define Long-Range and spell "ADCP'. Although I am puzzled by the fact that 'no real data is used for this paper' some details should be given on the HFR systems mentioned here.

two methods are introduced here and the abstract mentions that one seems to perform better than the other one - please provide quantitative information so to guide other users in their choice and critically assess the reasons why one method is performing better than the other.

Introduction. lines23-27: I don't understand this sentence. it seems to me that you are using horizontal interpolation (as described in the cited references) to reconstruct the vertical profile - which is not the case here. please rephrase this (and other sentences in the ms, possibly with the help of a native English speaking service- as most of the

sentences are long and convoluted and can be misinterpreted.

Section 2.2 Please provide quantitative figures of data reconstruction accuracy - even from different deployments as long as other readers have a clear idea of what we're aiming at here.

Skill assessment: this is done at a very basic level. there's plenty of good skill assessment approaches that would be more appropriate than what is used here.

Section 3.1. This needs to be rewritten in a more understandable way.

Section 3.2. define winter and summer seasons.

Overall, I think it has potential, but, I am puzzled and at this stage I am choosing to reconsidr after major revisions although I am leaning towards rejection. no real data is used -apart from the initialization of the covariance matrix, which should have been derived through HFR data instead. Using real data is complicated, fair enough, but this would guide users to a feasibility study in a more realistic scenario: what is the effect of data gaps, what is the data output rate that should be used (hourly-daily-weekly averaged HFR currents?). There is no discussion of the proposed approaches against data assimilation into the model, which has proven a very effective way of correcting a model's trajectory. there is no discussion of the computational requirements or efforts, again for instance against data assimilation into the models. If the proposed approaches are more effective (machine time - wise for instance) well that's would be beneficial indeed.

---

## Author Comment (AC1) · 17 Jan 2020

**Dear reviewer,**

**First, thank you for your careful review of our manuscript and your remarks. They have been really helpful to improve the manuscript and we have addressed them into the text, as explained in the point by point responses further down. Thank you for your specific comments on the paper structure as well, since they helped to realize that the explanation of the approach that we used was not clear enough. We hope that thanks to your suggestions we have managed to improve the manuscript, and that it suits now the standards of Ocean Science. Best regards,**

**Ivan Manso**

AR = Author's response

AC = Author's changes in the manuscript

All the changes' lines and pages correspond to the revised manuscript

**After considering the comments of the two anonymous referees, major changes have been made in the manuscript. First, we have better defined the context of this work using, among others, the references proposed by referee#2 in order to get a more complete introduction with regard to studies for the expansion of HFR data to subsurface levels. We have also changed the Sect. 3.1 into Sect. 3 separating it from the main results (now in Sect. 4), thus leaving its own section to the description of the simulated 'true' ocean. We have also clarified the main aim, approach and conclusions of our work, with changes in several parts of the manuscript which are detailed in the following point by point responses.**

Comments are enumerated

1- *Page 1, line 7: should technology be capitalized?*

    AR:  Done.

    AC: in page 1 line 8

2- *Page 1, line 16: should it read multiplatform or multisensory. Change to "aiming for the continuous".*

    AR: Both terms could be used,  but "multiplatform" is the term that better fits to the main focus of this paper, based on a model-based scenario where different platforms and sensors are measuring the same parameters and where different platforms are combined.

    AC: *"aiming for the continuous" c*orrected in the manuscript in page 1 lines 16-17.

3- *Page 1, line 17: change from "is today" to "are".*

AR: We have maintained "is" because it refers to the percentage, thus "is today" was changed by "is".

AC: "is today" was changed by "is" in page 1 line 17.

4- *Page 1, line 18: change to "resolution, but are limited to the".*

AR: We have rephrased the entire sentence and we have removed that part.

AC: rephased sentence in page 1 lines 17-18.

5- *The authors should make it more clear that the the ADCP and HFR data they are discussing were derived from model output. On the first read of the manuscript, I thought that data was from sensors deployed in the ocean.*

AR: The reviewer is right, and this was also the comment of referee #2. We have clarified this point with changes throughout the manuscript. In the new version, we explain that we use an assessment approach inspired by the techniques used in Observing System Simulation Experiments (OSSEs) where a numerical simulation is used as 'true ocean', which provides both, the observations and the 3D reference field that will be used to assess the results of the reconstruction (as shown in Fig. 2).

AC: There are changes in the abstract, introduction and Sec. 2.1 in order to better explain our main aim and the used approach. The title of the paper has also been changed to: "3D Reconstruction of Ocean Velocity from HFR and ADCP: a model-based assessment study", in order to make clearer this aspect of the methodology.

6- *Page 2, line 27: change to "combining simulated information from ".*

AR: We have changed the full sentence to make it clearer. In fact, we have fully changed this part of the Introduction.

7- *Page 2, line 30: change to "performance".*

AR: Done.

AC: in page 3, line 6

8- *Page 3, line 17: change to "with surface temperatures over".*

AR: Done

AC: in page 3, line 20

9- *Page 4, line 1: remove Moreover.*

AR: Done.

AC: in page 4, line 19

10- *Page 5, line 3: replace Summarizing with In summary,*

AR: Done.

AC: in page 5, line 18

11- *Page 5, line 11: change to "surface current fields along the Mid Atlantic".*

AR: Done.

AC: in page 5, line 26

12- *Page 6, line 19: can the authors be more specific on what is meant by the observations and the reference fields.*

AR: When dealing with methods for data 3D reconstruction, what we need to evaluate is the solution in the whole 3D domain, and namely in the areas that are not close to the observations. To this end, as explained in comment *5*, we use an assessment approach inspired by the techniques used in Observing System Simulation Experiments (OSSEs), where the observations that are used as inputs for the methods are emulated by a numerical simulation, and then the outputs (the reconstructed fields) are compared to the reference field obtained also from the 'true ocean' that is provided by such simulation. This approach is now better explained in Sect. 2.1 and different modifications through all the manuscript have been addressed accordingly.

13- *Page 7: What data source was used for the correlation scale tests, IBI, GLORYS-HR or GLORYS-LR?*

AR: The analysis of Sect. 3.1 (now changed to Sect. 3) provides an overview of the characteristics of the currents simulated by the numerical simulation from where the 'true' ocean was extracted. The IBI dataset was used for this purpose since as explained in this section, it has proven to be a realistic numerical simulation.
This change of Sect. 3.1 to Sect. 3 was made in order to make the manuscript clearer. In addition, note that the first paragraph of Sect. 2.3, where the numerical simulations are described, also links that section to this one.

AC: the initial configuration of the sections has been changed as mentioned, in addition to some changes throughout Sect. 3.

14- *Page 9: Figs 10-11 are mentioned before Figs 6-9, can this be changed.*

AR: Thank you, you are right.

AC: We have moved this paragraph to the end of the section as a general conclusion. Page 10, lines 20-24.

15- *Page 12, line 6: change to "the combination of synthetic data that mimics sensors from a multiplatform observing system to reconstruct".*

AR: We have changed the full paragraph to make clearer that we use emulated observations based on a realistic scenario as explained in the response to the comments before.

AC: in page 11

---

## Author Comment (AC2) · 17 Jan 2020

Dear reviewer,

Thank you for your thorough and critic review of our manuscript. Reading your comments, we have realized that neither the main idea nor the approach used to carry out our investigation were clear. Therefore, we have made significant changes throughout all the manuscript to correct these aspects and improve the readability of the paper. We are also very grateful for your suggestions on the literature cited in the paper and for providing additional context in terms of references. We hope that thanks to your suggestions we have managed to improve the manuscript, and that it suits now the standards of Ocean Science. The point by point response to your comments and the related changes are detailed in the following.

Best regards,

Ivan Manso

AR = Author's response

AC = Author's changes in the manuscript

All the changes' lines and pages correspond to the revised manuscript

**General response:**

AR = We deeply thank the reviewer for his/her comprehensive review of the manuscript and the very helpful comments. Indeed, we realized that our explanation on the approach used for the analysis of the methods' skills was not clear enough, and we have thoroughly rewritten several parts of the manuscript to correct this important aspect.

We agree with the reviewer that the most interesting aspect about the data-reconstruction methods is their application on real data. What we propose in this paper is a methodology for the evaluation of the 3D reconstruction methods prior to their application to real data. Our approach is based on the use of realistic numerical simulations as a 3D "true" ocean, that provides both, the observations and the 3D reference field that will be used to assess the results of the reconstruction ("reference field"). This is a well-established approach inspired by the techniques used in Observing System Simulation Experiments (OSSEs) and is the only approach that allows to test the methods in the full 3D domain considered for the reconstruction.

Performing a complete evaluation of the reconstructed fields would not be possible using only real observations, since it would be limited only to the areas covered by the observing systems. When dealing with methods for 3D data-reconstruction, what we need to evaluate is the solution in the whole domain, and namely in the areas that are not close to the observations.

In our opinion, this study is an essential prior step towards the applicability of this kind of methods on real data. Despite that the skill assessment approach is straightforward, it

provides the ground information and conclusions needed (i.e. where and why each method performs better/worse and their limitations) to know whether the application of the methods is feasible or not. We also think that the best-practice methodology developed in this work can be easily transferable to other locations and study cases, and prove very useful for expanding the use of 3D reconstruction on HFR observations.

The numerical simulation used as "true" ocean has proven to be realistic and in agreement with real data (in response with the reviewer's suggestions, this is now explicitly discussed in Sect. 3). We consider that the good agreement with the observations validates the approach used here and the conclusions obtained for our study area.

AC = **After considering the comments of the two anonymous referees, major changes have been made in the manuscript. First, we have better defined the context of this work using, among others, the references proposed by referee#2. The introduction of these references results in a more complete introduction, with regard to studies for the expansion of HFR data to subsurface levels. We have also changed the Sect. 3.1 into Sect. 3 separating it from the main results (now in Sect. 4); thus, leaving its own section to the description of the simulated "true" ocean. We have also clarified the main aim, approach and conclusions of our work, with changes in several parts of the manuscript which are detailed in the following point by point responses.**

**Point by point responses:**

Comments are enumerated.

1- *the title and the paper are misleading as they suggest that real data are used for the task, which unfortunately is not the case here.*

   AR: You are right, thanks for the comment. We have changed the title accordingly to provide a better idea on what it is done in this study.

   AC: the new title is "3D Reconstruction of Ocean Velocity from HFR and ADCP: a model-based assessment study"

2- *The manuscript indeed focuses on 'emulated' observations of currents provided through some 'supposedly' accurate and realistic model simulation. However, when it comes to the description of the model, the reader is pointed out to some references to other studies. If you have the model, why not compare that to the HFR data if you do not want to use the data itself for the task?*

   AR: First of all, sorry for not being clear with regard to the explanation of our main approach and the description of the model used to emulate the reality as "true" ocean (as explained in the general response). As mentioned before, to assess the results of the reconstruction we need a 3D "ground truth" that can be used to compare the different methods. And we need it with a good spatiotemporal coverage, which in practice can only be provided by numerical models. Then, it is important to be sure that the model is realistic enough, so the evaluations are meaningful. By realistic we mean that the model

has to reproduce the dominant processes in the region (i.e. permanent currents, mesoscale structures) so the spatiotemporal correlations among different locations are close to the actual ones.

In order to improve this aspect, a new section (Sect. 3) has been created from the former Sect 3.1, and it is used for assessing the realism of the simulations through their validation with previous studies based on observations in the study area (Rubio et al. (2013, 2019) and Solabarrieta et al. (2014)), .

AC: We have changed Sect. 3.1 into Sect. 3 (and rearranged the following sections accordingly) adding more discussion concerning the validation of the simulations. We have also added a paragraph in Sect. 2.3 (page 6 lines 20-22) to connect section 2.3 and the new Section 3.

3- *the literature review is lacking some important references. Development of subsurface current estimation procedures to complement surface currents started as soon as radar technologies were available. Some are given below, I leave the Authors to do a thoroughly review. Simple models dedicated to the prediction of current profiles have been developed (Prandle D., 1982. The vertical structure of tidal currents. Geophysical and Astrophysical Fluid Dynamics, 22, 29-49, 1982. Prandle D., 1987. The fine-structure of nearshore tidal and residual cirrculatins revealed by HF radar surface current measurements. Journal of Physical Oceanography, 17, 231-245, 1987. Prandle D., 1991. A view of near-shore dynamics based on observations from HF radar. Progress in Oceanography, 27, 403-438, 1991. ; Davies, 1982, 1983, 1985, 1992). Semi-empirical models, based on shallow-water hydrodynamics coupled to a modal representation of the current profiles, in which the modes have been estimated from local current profiles time series, have been used to estimate the 3-dimensional flow field from HF surface currents near the Rhine river outflow (de Valk C.F., 1999. Estimation of the 3-D current fields near the Rhine outflow from HF radar surface current data. Coastal Engineering, 37, 487-511, 1999.). A statistical method, based on vector correlation analysis between HF surface and ADCP subsurface currents and coupled with a modal representation in which modes were obtained from ADCP currents, was proposed in order to "project" surface currents along the water column. A different approach that infers the approximate shape of the current profiles from surface data without making use of local current profiles, has been introduced in 2001 for shallow-water coastal zone (Shen et Evans, 2001), subsequently extended to deep-water regions (Shen C.Y., Evans T., 2001. Surface-to-subsurface velocity projection for shallow water currents. Journal of Geophysical Research, 106, C4, 6973-6984, 2001. Shen C.Y., Evans T., 2002. Dynamically constrained projection for subsurface current velocity. Journal of Geophysical Research, 107, C11, 3203-3216, 2002. Doi: 10.1029/2001JC001036.), and is meant as an alternative to data assimilation into circulation models. The same approach has been recently applied to a shallow-water region in order to infer current profiles and to obtain maps of sea-surface slope from HF radar current estimates (Marmorino G.O., Shen C.Y., Evans T., Lindemann G.J., Hallock Z.R., Shay L.K., 2004. Use of 'velocity projection' to estimate the variation of sea-surface height from HF Doppler radar current measurements. Continental Shelf Research, 24, 353-374, 2004.). More recently, coupled with a two-layer density plume model, this technique was applied to estimate current profiles and density structure in a coastal zone dominated by a plume (Gangopadhyay A., Shen C.Y., Marmorino G.O., Mied R.P., Lindeman G.J., 2005. An extended velocity projection method for estimating the subsurface current and density structure for coastal plume regions: an application to the Chesapeake Bay outflow). The so-called "Velocity Projection Technique" introduced in these papers, relies on the surface-to- subsurface viscous coupling and turbulent transfer of momentum and shear in order to*

*infer the velocity distribution over depth from measured surface currents and wind stress. This method, applied in its original formulation to shallow coastal water, resolves the vertical structure of the currents in terms of a finite expansion of orthogonal modes spanning the water column. The modal weights are obtained by applying appropriate dynamical constraints to the inferred current profiles and their vertical derivatives at the boundaries.*

AR: Thank you very much for providing additional context in term of references about studies that investigate procedures to expand surface HFR information to subsurface levels. We have rewritten part of the Introduction accordingly.

AC: We have changed part of the introduction (page 2, line 24 – page 3, line 2):
"In the last years, several methods to expand the information of the HFR data to subsurface layers in the upper water column have been developed, such as: the use of multifrequency radars to obtain the velocity shear (Stewart and Joy, 1974; Barrick, 1972; Broche et al., 1987; Paduan and Graber, 1997; Teague et al., 2001), the use of the secondary peaks in the radar echo spectra to obtain the velocity shear (Shrira et al., 2001; Ivonin et al., 2004) or the "velocity projection" method to obtain the velocities of the subsurface currents (Shen and Evans, 2002; Marmonio et al., 2004; Gangopadhyay et al., 2005). Besides, simple models that study the surface and vertical profiles have been developed (e.g. Prandle, 1982, 1987, 1991; Davies, 1985a, 1985b, 1985c). In addition, other approaches combine the HFR data with data in the water column provided by in-situ moored instruments, remote sensing platforms or circulation numerical simulations to investigate the 3D circulation (e.g. C.F. de Valk, 1999; O'Donncha et al., 2014; Cianelli et al., 2015; Ren et al., 2015; Jordà et al., 2016; Fredj, 2016)."

4- *Abstract / main body: define surface. HFR sense different 'depths' based on the working frequency. Define Long-Range and spell "ADCP'. Although I am puzzled by the fact that 'no real data is used for this paper' some details should be given on the HFR systems mentioned here.*

AR: Modified.

AC: Surface defined in page 2 line 19. ADCP spelled in page 3 line 6. Details of the HFR used to define the numerical-based scenario for this work are now given in page 3 line 23.

5- *two methods are introduced here and the abstract mentions that one seems to perform better than the other one - please provide quantitative information so to guide other users in their choice and critically assess the reasons why one method is performing better than the other.*

AR: In the abstract it is mentioned that in general the methods perform better in well sampled areas and that both show different performances. However, it was not our aim to say that one is better than the other. In order to make it clearer we have removed the phrase where we said that different performances were observed between them.
Actually, as explained in the conclusions section, each method has its pros and cons and it is difficult to summarize them in the abstract. In Sect. 4.1, where the results are presented, it is shown that the DCT-PLS performs better in well sampled areas whereas the ROOI performs better in the rest of the areas, although it also performs well in well sampled areas.

AC: We have removed the phrase "although different performances between the methods are observed". The whole abstract has been revised to improve clarity on the aim, methods and results of our work.

6- *Introduction. lines23-27: I don't understand this sentence. it seems to me that you are using horizontal interpolation (as described in the cited references) to reconstruct the vertical profile - which is not the case here. please rephrase this (and other sentences in the ms, possibly with the help of a native English speaking service- as most of the sentences are long and convoluted and can be misinterpreted.*

AR: We agree with that this sentence was confusing. We have changed the part of the introduction where the methods for inferring sub-surface currents are described and where the methods that we use are introduced. We hope that now it is clearer. English language has been reviewed through all the manuscript, sentences have been shortened and simplified when needed.

AC: The main change in the introduction is made between page 2, line 24- page 3, line 10.

7- *Sect. 2.2 Please provide quantitative figures of data reconstruction accuracy – even from different deployments as long as other readers have a clear idea of what we're aiming at here.*

AR: Thank you, this is an interesting point. However, it is not easy to compare the cases of those applications with our case since the data, study area and configurations were different. Also, same RMSD, for instance, can have a very different meaning depending on the characteristics of the circulation in each region. In any case if the reader is interested in those results, they can now be found in the references.

8- *Skill assessment: this is done at a very basic level. there's plenty of good skill assessment approaches that would be more appropriate than what is used here.*

AR: We agree that there are plenty options of the skill assessment of the methods, both from the eulerian or the lagrangian perspective. However, it is also true that the RMSD (and relative RMSD) is an intuitive quantity that provide a first evaluation of the error in the same units as the variable being evaluated (or as a percent to its variability). Other complementary diagnostics could have been used (and were tried in a previous version of the manuscript), but they do not provide much further insight while they make the manuscript denser. Therefore, we prefer to stick to the chosen diagnostics that have proven enough to evaluate the benefits and drawbacks of each method. In any case, we put special effort in the computation of the RMSD in different ways by using maps and spatial and temporal means with the aim to characterise in a comprehensive way the 3D structure of the errors associated to the reconstructed fields.

9- *Sect. 3.1. This needs to be rewritten in a more understandable way.*

AR: Thank you. The whole section has been revised. Sect. 3.1 is now Sect. 3 and sections Sect. 2.4 and 3 are now better connected. We hope that these changes have made this section clearer. Part of the contents have been moved to an appendix in order to ease the reading.

AC: the initial configuration of the sections has been changed as mentioned, in addition to some changes to the main text, now in Sect. 3.

10- *Sect. 3.2. define winter and summer seasons.*

AR: They are already defined in Sect. 2.1: "winter and summer periods: Nov-Dec-Jan-Feb (2010-2011) and Jun-Jul-Aug-Sep (2011), respectively".

11- *no real data is used -apart from the initialization of the covariance matrix, which should have been derived through HFR data instead.*

AR: For the application of the ROOI method we needed to define a 3D velocity covariance matrix (eq. 1). To do this, time series of velocity in each grid point where the reconstruction will be performed, are required. This cannot be achieved from real data, so this matrix was obtained from a realistic model simulation. Note that here the hypothesis is that the statistical relationships between different locations are well captured by the model, which is less demanding than requiring that the velocity field is well reproduced at each time step. Moreover, we have tested the impact of the choice of different numerical simulations for the computation of the 3D velocity covariance matrix (IBI, GLORYS-HR and GLORYS-LR, see Figs 10-11). The goal of this test is to see what would happen if the model used did not totally represent the same dynamics than reality (i.e. the modelled covariance matrix was inaccurate).

12- *Using real data is complicated, fair enough, but this would guide users to a feasibility study in a more realistic scenario: what is the effect of data gaps, what is the data output rate that should be used (hourly-daily-weekly averaged HFR currents?).*

AR: As mentioned before, we do not use real data because 3D reconstructions could not be validated with existing data (i.e. data sampling in subsurface is very scarce). Thus, we work in the "model world" to better identify limitations and skills of the reconstruction methods. However, the reviewer is right in that we could have used a more realistic configuration for instance by adding gaps and errors to the emulated data. Unfortunately, the paper is already long, so we cannot include more sensitivity experiments. Moreover, this has already been addressed in a 2D context by other authors (* e.g. Hernández-Carrasco, 2018; Kaplan and Lekien, 2007; Yaremchuk and Sentchev, 2009). We added a sentence explaining that it would be an interesting further step to evaluate the robustness of the reconstruction methods when dealing with observational issues.

With regard to the data input rate, for this study hourly data were used. While not limitations on the time resolution exist for the DCT-PLS, the ROOI outputs will be limited to the temporal resolution of the covariance matrix. In a case with real data, the temporal resolution of the reconstruction should be chosen in coherence to the process to be monitored/studied and after examination of the spatio-temporal correlations.

*Hernández-Carrasco, I., Solabarrieta, L., Rubio, A., Esnaola, G., Reyes, E., and Orfila, A.: Impact of HF radar current gap filling methodologies on the Lagrangian assessment of coastal dynamics, Ocean Sci., 14, 827-847, https://doi.org/10.5194/os-14-827-2018, 2018.

Kaplan, D. M., and Lekien, F.: Spatial interpolation and filtering of surface current data based on open-boundary modal analysis, J. Geophys. Res.-Oceans, 112, C12007, https://doi.org/10.1029/2006JC003984, 2007.

Yaremchuk, M., and Sentchev, A.: Mapping radar-derived sea surface currents with a variational method, Cont. Shelf Res., 29, 1711-1722, https://doi.org/10.1016/j.csr.2009.05.016, 2009.

AC: We added a sentence explaining that it would be an interesting further step to evaluate the robustness of the reconstruction methods for different observational errors in page 11 lines 20-21.

13- *There is no discussion of the proposed approaches against data assimilation into the model, which has proven a very effective way of correcting a model's trajectory.*
*there is no discussion of the computational requirements or efforts, again for instance against data assimilation into the models. If the proposed approaches are more effective (machine time - wise for instance) well that's would be beneficial indeed.*

AR: We thank the reviewer for this suggestion that is useful to clarify the scope of the methodologies. Indeed, the ROOI method shares several characteristics with what is used in data assimilation (DA) methods, in particular in sequential methods like Optimal Interpolation (* Ide et al., 1997) or the SEEK filter (* Brasseur and Verron, 2006). So, the comment is very appropriate.
The main difference between this type of reconstructions and data assimilation is not in the methodology *per se* but in the overall approach. Data assimilation is typically applied to correct a model trajectory. That is, the background field used is a model forecast, that is combined with the information from observations to create an analysed field. This field is then used to initialize the model for the next simulation cycle. In the reconstruction methods we analyse in this work, the 3D velocity field is inferred solely from the observations. Only in the case of ROOI a model is used to provide the covariance matrices, which are fixed in time (i.e. only computed once).
In summary the main differences of these methods with DA are (1) no model has to be run to provide a background field, so the procedure is much faster and (2) most of the information is provided by the observations and the model is only used in the ROOI case to provide background statistics.

* Ide, K., Courtier, P., Ghil, M., and Lorenc, A. C.: Unified Notation for Data Assimilation: Operational, Sequential and Variational (gtSpecial IssueItData Assimilation in Meteology and Oceanography: Theory and Practice). *Journal of the Meteorological Society of Japan. Ser. II*, *75*(1B), 181-189, 1997.

*Brasseur, P., and Verron, J.: The SEEK filter method for data assimilation in oceanography: a synthesis. *Ocean Dynamics*, *56*(5-6), 650-661, 2006.

AC: We have added a sentence showing the added values of the methods compared to data assimilation in the conclusions in page 12 lines 6-8

---

## Referee Report (RR1)

The previous referees did an excellent job reviewing the article and highlighting the areas that needed changing, clarifying or correcting.
It is my opinion that the authors responded adequately to the reviewers' comments, making major changes in the manuscript where needed, clarifying vague parts, and also answered the many and well placed reviewers' questions.

A couple of minor typos I noticed and should be corrected:

page 1, line 17: replace "percent" by "percentage"
page 1, line 21: with complementary data**, such** as
page 2, line 18: replace "relative to" with "at"

Concluding, I believe the final version of the manuscript is adequately well written and has the required scientific value for publication.

---

## Author Response (AR2)

**AUTHOR'S RESPONSE**

**Topic Editor's comments**

The previous referees did an excellent job reviewing the article and highlighting the areas that needed changing, clarifying or correcting.

It is my opinion that the authors responded adequately to the reviewers' comments, making major changes in the manuscript where needed, clarifying vague parts, and also answered the many and well placed reviewers' questions.

A couple of minor typos I noticed and should be corrected:
page 1, line 17: replace "percent" by "percentage"
page 1, line 21: with complementary data, such as
page 2, line 18: replace "relative to" with "at"

Concluding, I believe the final version of the manuscript is adequately well written and has the required scientific value for publication.

**Author's response**

**Dear George Petihakis,**

**Thank you for accepting the paper and for the comments about the typos. We have already corrected them.**

**Best regards,**
**Ivan Manso**

Marked-up version of the manuscript:

[revised manuscript text omitted]